# General decapping activators target different subsets of inefficiently translated mRNAs

**Feng He[†], Alper Celik[†‡], Chan Wu, Allan Jacobson***

Department of Microbiology and Physiological Systems, University of Massachusetts Medical School, Massachusetts, United States

**Abstract** The Dcp1-Dcp2 decapping enzyme and the decapping activators Pat1, Dhh1, and Lsm1 regulate mRNA decapping, but their mechanistic integration is unknown. We analyzed the gene expression consequences of deleting *PAT1, LSM1,* or *DHH1,* or the *DCP2* C-terminal domain, and found that: i) the Dcp2 C-terminal domain is an effector of both negative and positive regulation; ii) rather than being global activators of decapping, Pat1, Lsm1, and Dhh1 directly target specific subsets of yeast mRNAs and loss of the functions of each of these factors has substantial indirect consequences for genome-wide mRNA expression; and iii) transcripts targeted by Pat1, Lsm1, and Dhh1 exhibit only partial overlap, are generally translated inefficiently, and, as expected, are targeted to decapping-dependent decay. Our results define the roles of Pat1, Lsm1, and Dhh1 in decapping of general mRNAs and suggest that these factors may monitor mRNA translation and target unique features of individual mRNAs.

DOI: https://doi.org/10.7554/eLife.34409.001

**\*For correspondence:**
Allan.Jacobson@umassmed.edu

[†]These authors contributed equally to this work

**Present address:** [‡]The Centre for Computational Medicine, Peter Gilgan Centre for Research and Learning, Toronto, Canada

**Competing interests:** The authors declare that no competing interests exist.

## Introduction

Decapping usually commits an mRNA to complete degradation and plays an important role in eukaryotic cytoplasmic mRNA turnover (*Valkov et al., 2017*; *Grudzien-Nogalska and Kiledjian, 2017*; *Parker, 2012*). Decapping is required for general 5' to 3' mRNA decay (*Decker and Parker, 1993*), nonsense-mediated mRNA decay (NMD) (*He and Jacobson, 2001*), AU-rich element-mediated mRNA decay (*Yamashita et al., 2005*; *Fenger-Grøn et al., 2005*), microRNA-mediated gene silencing (*Behm-Ansmant et al., 2006*), and transcript-specific degradation (*Dong et al., 2007*; *Badis et al., 2004*). In yeast, mRNA decapping is carried out by a single enzyme comprised of a regulatory subunit (Dcp1) and a catalytic subunit (Dcp2). Dcp1 is a small EVH domain protein essential for mRNA decapping in vivo (*She et al., 2004*; *Beelman et al., 1996*) and Dcp2 is a 970-amino acid protein containing a highly conserved Nudix domain at its N-terminus and a large extension at its C-terminus (*Gaudon et al., 1999*; *Dunckley, 1999*). The Dcp2 N-terminal domain is essential for the catalysis of cap removal, but may also have additional regulatory activity, as this domain also contains the binding site for Dcp1 (*She et al., 2008*; *Deshmukh et al., 2008*) and interacts with the activation motif of a short Edc1 peptide in vitro in the presence of Dcp1 (*Mugridge et al., 2018*; *Valkov et al., 2016*). The decapping role of the Dcp2 C-terminal domain is largely unknown. However, our recent experiments reveal that this domain includes both negative and positive regulatory elements that control both the substrate specificity and the activation of the decapping enzyme (*He and Jacobson, 2015a*).

In addition to the Dcp1-Dcp2 decapping enzyme, mRNA decapping also requires the functions of specific regulators commonly dubbed 'decapping activators' (*Parker, 2012*). A large number of decapping activators have been identified in yeast and other organisms (*Jonas and Izaurralde, 2013*; *Parker, 2012*), and these factors appear to target distinct classes of mRNA substrates. Pat1,

Dhh1, and the Lsm1-7 complex are required for decapping of general wild-type mRNAs (*Parker, 2012*), and NMD-specific regulators (Upf1, Upf2, and Upf3) are required for decapping of nonsense-containing mRNAs (*He and Jacobson, 2015b*; *Nicholson and Mühlemann, 2010*). Edc3 manifests the most fastidious substrate specificity, being required for decapping of only the yeast *YRA1* pre-mRNA and *RPS28B* mRNA (*He et al., 2014*; *Dong et al., 2007*; *Badis et al., 2004*). Two additional factors, Edc1 and Edc2, originally isolated as high-copy suppressors of specific mutations in *DCP1* and *DCP2*, by themselves appear to be not required for mRNA decapping in vivo but can enhance mRNA decapping in vitro (*Borja et al., 2011*; *Dunckley et al., 2001*). All of these decapping activators are conserved from yeast to humans, but their precise functions in mRNA decapping regulation are largely unknown. The two major functions proposed for yeast decapping activators, translational repression and decapping enzyme activation (*Parker, 2012*; *Nissan et al., 2010*; *Coller and Parker, 2005*), are still controversial (*Sweet et al., 2012*; *Arribere et al., 2011*).

Yeast decapping activators exhibit highly specific interactions with each other and with the decapping enzyme. Pat1 interacts with both Dhh1 and the Lsm1-7 complex (*He and Jacobson, 2015a*; *Sharif et al., 2013*; *Sharif and Conti, 2013*; *Nissan et al., 2010*; *Bouveret et al., 2000*; *Wu et al., 2014*), Upf1, Upf2, and Upf3 interact with each other (*He et al., 1997*), and Edc3 interacts with Dhh1 (*He and Jacobson, 2015a*; *Sharif et al., 2013*). Pat1, Upf1, and Edc3 also interact with specific binding motifs in the large C-terminal domain of Dcp2 (*He and Jacobson, 2015a*; *Harigaya et al., 2010*). These interaction data and additional observations led us to propose a new model for regulation of mRNA decapping (*He and Jacobson, 2015a*) in which different decapping activators form distinct decapping complexes in vivo, each of which has a unique substrate specificity that targets a subset of yeast mRNAs. To test aspects of this model, and to further understand the roles of Pat1, Dhh1, and the Lsm1-7 complex in general mRNA decapping we have analyzed the effects of deletions of the *PAT1*, *LSM1*, or *DHH1* genes and the large Dcp2 C-terminal domain on transcriptome-wide mRNA accumulation. Our results reveal a critical role for the Dcp2 C-terminal domain in regulating mRNA decapping, demonstrate that Pat1, Lsm1, and Dhh1 control the decapping of specific subsets of yeast mRNAs, and uncover substantial indirect consequences of mutations in genes encoding components of the decapping apparatus.

## Results

### Elimination of the large Dcp2 C-terminal domain causes significant changes in genome-wide mRNA expression

We previously identified multiple regulatory elements in the large C-terminal domain of Dcp2, including one negative element that inhibits in vivo decapping activity and a set of positive elements that promote both substrate specificity and decapping activation. The latter appear to operate by binding to specific decapping activators such as Upf1, Edc3, and Pat1 (*He and Jacobson, 2015a*). To extend our previous study and to further assess the roles of these Dcp2 regulatory elements in in vivo decapping control, we analyzed the effect of C-terminal truncation of Dcp2 on transcriptome-wide mRNA accumulation. RNA-Seq was used to analyze transcript populations in wild-type yeast cells and in cells harboring the previously characterized *dcp2-N245* allele (*He and Jacobson, 2015a*). This allele produces a Dcp2 decapping enzyme that contains only the first 245 amino acids of the protein, and appears to have constitutively activated and indiscriminate decapping activity, at least with respect to the limited number of mRNAs analyzed previously (*He and Jacobson, 2015a*). As with any mutation, truncation of Dcp2 could have a direct effect on mRNA decapping, or it could have an indirect effect on overall gene expression. To identify those transcripts directly affected by C-terminal truncation of Dcp2, we constructed two additional isogenic strains with severely compromised decapping activity and included these two strains in our RNA-Seq experiments. These strains, dubbed *dcp2-E153Q-N245* and *dcp2-E198Q-N245*, harbor the same *dcp2-N245* allele but each also contains one additional function-inactivating mutation in an active site residue of the Dcp2 Nudix domain, that is glutamate (E) to glutamine (Q) substitutions at codon positions 153 and 198, respectively. E153 of Dcp2 has been shown to function as a general base during the hydrolysis reaction and E198 is involved in $Mg^{2+}$ coordination within the Nudix domain (*Aglietti et al., 2013*). *E153Q* and *E198Q* mutations essentially eliminate the decapping activity of Dcp2 both in vitro and in vivo (*Aglietti et al., 2013*; *He and Jacobson, 2015a*).

RNA-Seq libraries prepared from wild-type cells and from the isogenic strains harboring the *dcp2-N245, dcp2-E153Q-N245,* or *dcp2-E198Q-N245* alleles showed good read count distribution (*Figure 1A*) and notable consistency between biological replicates, with Pearson correlation coefficients ranging from 0.96 to 0.99 (*Figure 1* and *Figure 1—figure supplement 1*). Utilizing previously described data analysis pipelines for transcript quantitation and assessment of differential expression (*Celik et al., 2017*), we identified the transcripts that were differentially expressed in each of the mutant strains relative to the wild-type strain. The decapping-deficient *dcp2-E153Q-N245* and *dcp2-E198Q-N245* strains exhibited significant numbers of transcripts that were differentially expressed. We identified 1921 up-regulated and 1845 down-regulated transcripts in the *dcp2-E153Q-N245* strain, and 1346 up-regulated and 1428 down-regulated transcripts in the *dcp2-E198Q-N245* strain (*Figure 1B,C,D*). Given the general requirement for the decapping enzyme in yeast mRNA decay (*Parker, 2012*), the detection of a large number of up-regulated transcripts in these two strains was not surprising, that is the up-regulated transcripts are most likely *bona fide* substrates of the yeast decapping enzyme. In support of this interpretation, the up-regulated transcript lists from these strains contain all our previously characterized individual decapping substrates and also exhibited highly significant overlap with the up-regulated transcript lists from *dcp1Δ, dcp2Δ, xrn1Δ,* and *upf1/ 2/3Δ* cells (*Figure 1—figure supplement 2*). In contrast, the finding that a large number of transcripts was also down-regulated in the two decapping-inactive strains was surprising. Similar observations were also made in our recent RNA-Seq analyses of *dcp1Δ* and *dcp2Δ* cells (*Celik et al., 2017*). These results indicate that general inhibition of mRNA decapping may also have severe secondary effects on transcriptome-wide mRNA accumulation.

Examination of the up- and down-regulated transcript lists from the two catalytically inactive strains revealed a significant overlap, but also notable differences (*Figure 1B,C*). The two strains share 1186 up-regulated and 1362 down-regulated transcripts. However, the *dcp2-E153Q-N245* strain yielded 575 more up-regulated and 417 more down-regulated transcripts than the *dcp2-E198Q-N245* strain. To explore whether there was a significant difference in the expression patterns between these two strains, we applied the same differential expression pipeline but compared the *dcp2-E153Q-N245* and *dcp2-E198Q-N245* libraries directly. This analysis revealed only 21 differentially expressed transcripts between these two strains (*Figure 1E*, leftmost panel). From this result, we conclude that there is no fundamental difference in expression patterns between the two strains. However, because the two strains harbor different *dcp2* alleles, the encoded decapping enzymes may have slightly different albeit significantly reduced activities. Thus, some subtle differences in levels of expression may actually exist for a large number of transcripts between the two strains, as we noticed in our validation experiments (see below). The subtle differences in levels of expression for these transcripts were likely captured in the *dcp2-E153Q-N245* vs. *WT* but not in the *dcp2-E198Q-N245* vs. *WT* comparison.

Elimination of the entire C-terminal domain of Dcp2 significantly altered genome-wide mRNA expression. Compared to WT cells, a total of 1530 transcripts were differentially expressed in *dcp2-N245* cells: 616 transcripts showed up-regulation and 914 transcripts showed down-regulation (*Figure 1B,C*). To assess the specific effect of the C-terminal truncation of Dcp2 on mRNA decapping, we compared the expression pattern of the *dcp2-N245* strain to the patterns of the *dcp2-E153Q-N245* and *dcp2-E198Q-N245* strains (*Figure 1B,C*). Transcripts differentially expressed in these three strains shared a partial overlap, but also exhibited notable differences. A significant fraction of transcripts differentially expressed in the *dcp2-N245* strain (324 out of 616 for the up-regulated, and 650 out 914 for the down-regulated) had concordant up- or down-regulation in the catalytically inactive strains. Interestingly, the majority of transcripts differentially expressed in the catalytically inactive strains (1567 out of 1891 for the up-regulated, and 1261 out of 1911 for the down-regulated) had unchanged levels in the *dcp2-N245* strain. Furthermore, a significant fraction of transcripts differentially expressed in the *dcp2-N245* strain (292 out 616 for the up-regulated, and 264 out of 914 for the down-regulated) had unchanged levels in the catalytically inactive strains. Together, these results indicate that the *dcp2-N245* strain and the two catalytically inactive strains have largely different global mRNA expression patterns and suggest that the C-terminal truncation of Dcp2 does not block general mRNA decapping, but causes deregulation of decapping for specific mRNAs. To further support this conclusion, we carried direct pairwise comparisons between the *dcp2-N245* and the *dcp2-E153Q-N245* or *dcp2-E198Q-N245* libraries. The *dcp2-N245* strain yielded 1658 up-regulated and 1690 down-regulated transcripts compared to the *dcp2-E153Q-N245* strain,

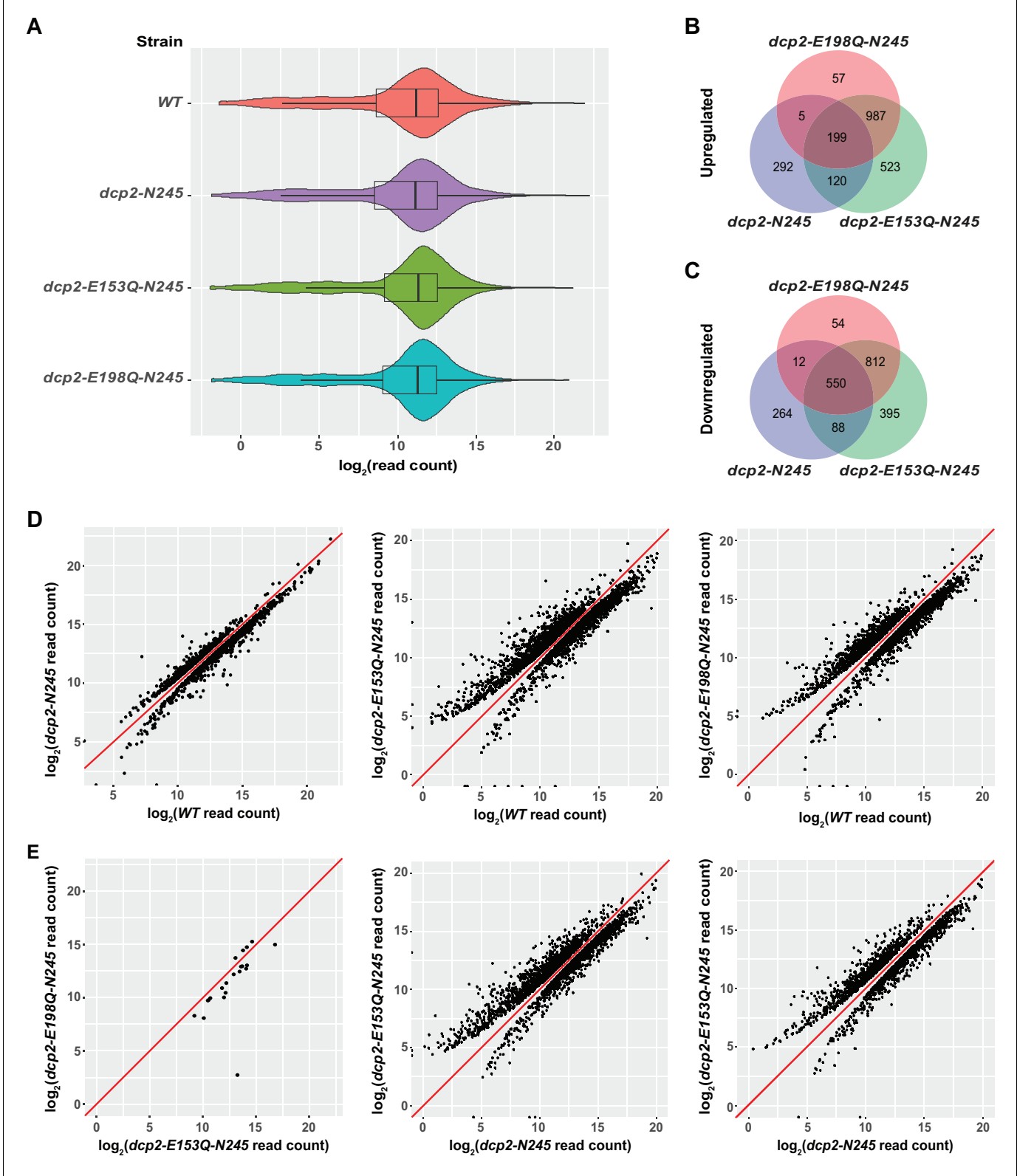

**Figure 1.** Identification of transcripts differentially expressed in *dcp2-N245, dcp2-E153Q-N245*, and *dcp2-E198Q-N245* cells. (A) Violin and box plots displaying the average and median read count distributions of the RNA-Seq libraries from *WT, dcp2-N245, dcp2-E153Q-N245*, and *dcp2-E198Q-N245* strains in three independent experiments. (B) Venn diagram displaying the relationships between transcripts up-regulated in *dcp2-N245, dcp2-E153Q-N245*, and *dcp2-E198Q-N245* cells. (C) Venn diagram displaying the relationships between transcripts down-regulated in *dcp2-N245, dcp2-E153Q-*

*Figure 1 continued*

N245, and *dcp2-E198Q-N245* cells. (D) Scatterplots comparing the normalized read counts between the *WT* and the *dcp2-N245, dcp2-E153Q-N245,* or *dcp2-E198Q-N245* strains for transcripts differentially expressed in each of the mutant strains. Left panel, comparison for the 616 up- and 1025 down-regulated transcripts in *dcp2-N245* cells; middle panel, comparison for the 1921 up- and 1845 down-regulated transcripts in *dcp2-E153Q-N245* cells; and right panel, comparison for the 1346 up- and 1428 down-regulated transcripts in *dcp2-E198Q-N245.* (E) Scatterplots comparing the normalized read counts for transcripts differentially expressed between the *dcp2-E153Q-N245* and *dcp2-E198Q-N245* strains, or in these two strains compared to the *dcp-N245* strain. Left panel, comparison for 21 differentially expressed transcripts between *dcp2-E153Q-N245* and *dcp2-E198Q-N245* cells; middle panel, comparison for the 1658 up- and 1690 down-regulated transcripts in *dcp2-E153Q-N245* cells; and right panel, comparison for the 1113 up- and 1090 down-regulated transcripts in *dcp2-E198Q-N245* cells. The $\log_2$ read count values of individual transcripts were used in the analyses of parts D and E, and the y = x line is shown in red.

DOI: https://doi.org/10.7554/eLife.34409.002

The following figure supplements are available for figure 1:

**Figure supplement 1.** RNA-Seq libraries generated from *WT, dcp2-N245, dcp2-E153Q-N245,* and *dcp2-E198Q-N245* strains exhibit good correlation between three different biological replicates.

DOI: https://doi.org/10.7554/eLife.34409.003

**Figure supplement 2.** Yeast transcripts stabilized by inactivating the catalytic function of Dcp2 are mostly decapping substrates.

DOI: https://doi.org/10.7554/eLife.34409.004

**Figure supplement 3.** Yeast transcripts destabilized by deletion of the large Dcp2 C-terminal domain are not normally typical decapping substrates.

DOI: https://doi.org/10.7554/eLife.34409.005

**Figure supplement 4.** Western blotting analysis of protein levels in cells with different *dcp2* alleles.

DOI: https://doi.org/10.7554/eLife.34409.006

and 1113 up-regulated 1090 down-regulated compared to the *dcp2-E198Q-N245* strain (**Figure 1E**, middle and right panels), further illustrating these differences. The distinct mRNA expression patterns observed in *dcp2-N245, dcp2-E153Q-N245*, and *dcp2-E198Q-N245* cells most likely originated from different Dcp2 decapping activities, as the levels of the Dcp2 proteins expressed in these cells were similar or nearly identical (**Figure 1—figure supplement 4**).

## Elimination of the C-terminal domain of Dcp2 deregulates but does not block mRNA decapping

Our comparison of the transcripts differentially expressed between *dcp2-N245* cells and cells expressing the two decapping-deficient alleles suggested that the *dcp2-N245* truncation causes deregulated decapping but not decapping inhibition. To explore this concept further, we examined correlations of the transcriptome-wide profiles of all transcripts in the *dcp2-N245* strain and in yeast strains severely comprised in decapping activity (*dcp2-E153Q-N245* or *dcp2-E198Q-N245* cells) or strains lacking decapping or 5' to 3' exoribonuclease activities (*dcp1Δ, dcp2Δ,* and *xrn1Δ* cells). As controls, we also did pairwise comparisons of the profiles for the latter group of strains. In this analysis, profiling data for the *dcp1Δ, dcp2Δ,* and *xrn1Δ* strains were from our recently published study (**Celik et al., 2017**). Because the *dcp1Δ, dcp2Δ,* and *xrn1Δ* libraries were prepared at a different time, to improve the consistency, we used the relative levels (i.e., the fold changes relative to the corresponding *WT* control) of each transcript in all these strains in our analyses. As shown in **Figure 2A**, the *dcp1Δ* and *dcp2Δ* strains, or the *dcp2-E153Q-N245* and *dcp2-E198Q-N245* strains, showed excellent correlation (Pearson correlation coefficients = 0.868 and 0.932, respectively). The *dcp1Δ* and *dcp2Δ* strains also showed good correlation with the *dcp2-E153Q-N245, dcp2-E198Q-N245* or *xrn1Δ* strains (Pearson correlation coefficients = 0.812, 0.805, 0.748 for *dcp1Δ* and 0.772, 0.789, 0.734 for *dcp2Δ* strains, respectively). In contrast, the *dcp2-N245* strain exhibited only modest correlation with each of the *dcp1Δ, dcp2Δ, dcp2-E153Q-N245, dcp2-E198Q-N245,* or *xrn1Δ* strains (Pearson correlation coefficients = 0.371, 0.345, 0.423, 0.421, and 0.379). These results indicate that the *dcp2-N245* strain has a significantly different expression profile from yeast strains severely deficient in or lacking decapping or 5' to 3' exoribonuclease activities, further arguing that the C-terminal truncation of Dcp2 deregulates but does not block mRNA decapping.

To validate our RNA-Seq results and to assess the potential mechanisms of decapping deregulation caused by elimination of the Dcp2 C-terminal domain, we focused on a group of 264 transcripts that were down-regulated uniquely in *dcp2-N245* cells (**Figure 1C**). Because the levels of transcripts from this group were not altered in *dcp2-E153Q-N245* or *dcp2-E198Q-N245* cells, we reasoned that

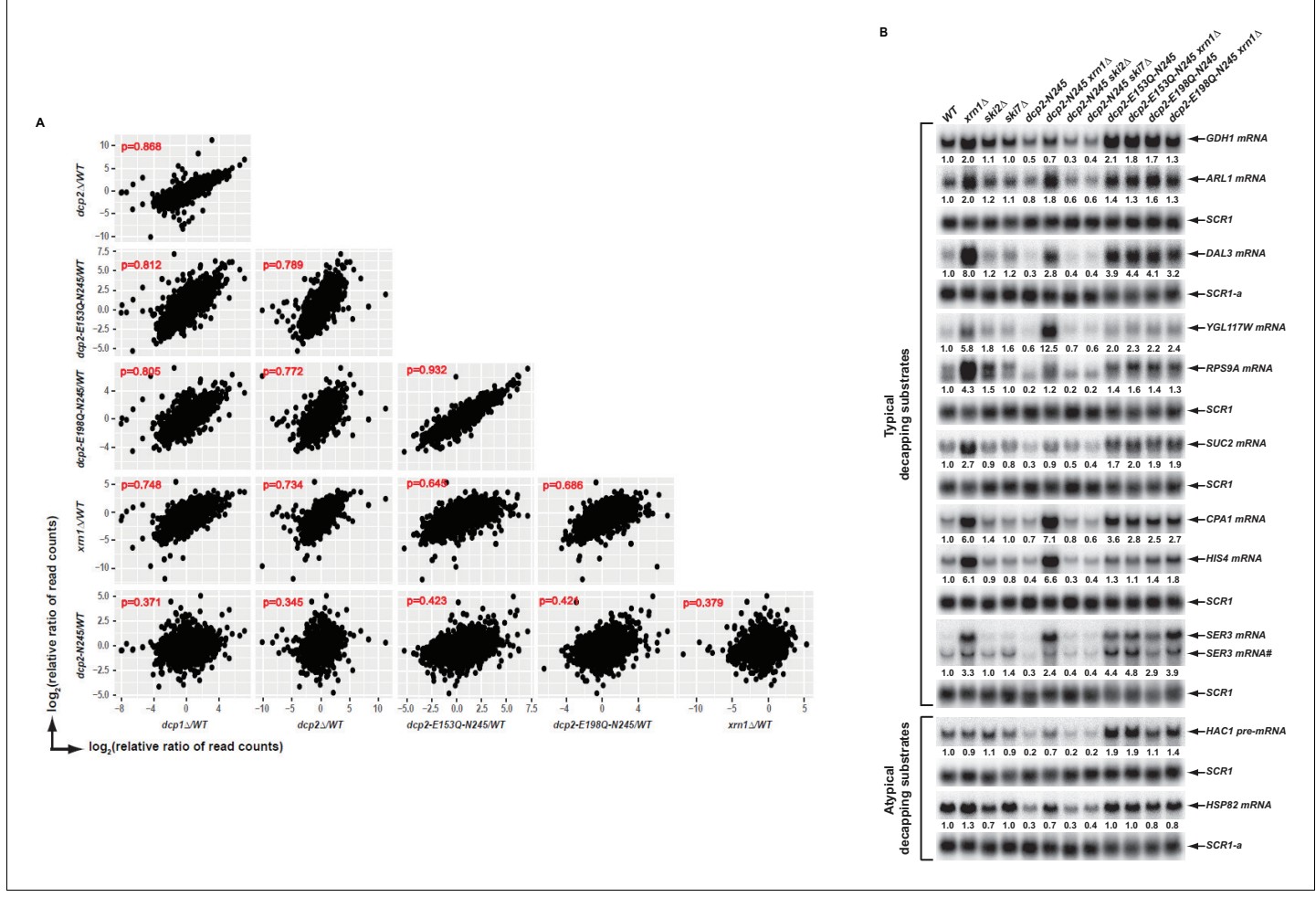

**Figure 2.** Elimination of the Dcp2 C-terminal domain deregulates mRNA decapping in vivo. (**A**) Yeast cells harboring a deletion of the large Dcp2 C-terminal domain exhibit a significantly different genome-wide expression pattern from cells severely comprised in decapping activity or completely lacking decapping or 5' to 3' exoribonuclease activities. Scatterplot matrices were used to compare the relative levels of all transcripts in the yeast transcriptome in different mutant strains. The relative levels of individual mRNAs in each of the mutant strains were determined by comparisons to the appropriate wild-type strain. Data for the *dcp1Δ*, *dcp2Δ*, and *xrn1Δ* strains were from our previous study (*Celik et al., 2017*). Log2 transformed data were used for this analysis and Pearson correlation coefficients for each comparison are shown in red. (**B**) Yeast cells harboring a deletion of the large Dcp2 C-terminal domain exhibit accelerated and indiscriminate decapping of mRNAs. Eleven representative transcripts (nine typical decapping substrates and two atypical decapping substrates) from the group of transcripts down-regulated uniquely in *dcp2-N245* cells were selected and their levels of expression in the indicated strains were analyzed by northern blotting. In each case, a specific random-primed probe was hybridized to the blot and the *SCR1* transcript served as a loading control. The relative levels of specific transcripts in the mutant strains were determined by comparisons to their levels in the wild-type strain (indicated by the values under each blot). For presentation purposes, one of the control *SCR1* blots is duplicated and is indicated by the lower case letter 'a.' The *SER3* locus produces two different transcripts and only the levels of the short isoform (indicated by #) are presented.

DOI: https://doi.org/10.7554/eLife.34409.007

their decreased accumulation in *dcp2-N245* cells was likely to be caused by accelerated or indiscriminate decapping by a constitutively activated Dcp2 decapping enzyme which had lost its negative regulation. To test this hypothesis, we examined whether elimination of *XRN1*, a gene encoding the 5' to 3' exoribonuclease that functions downstream of decapping, can restore the levels of the down-regulated transcripts in *dcp2-N245* cells. To assess the specificity of *XRN1* deletion, we also analyzed the effects of elimination of *SKI2* or *SKI7* on the accumulation of the down-regulated transcripts in *dcp2-N245* cells. *SKI2* encodes a 3' to 5' RNA helicase, *SKI7* encodes a GTPase, and both gene products are required for exosome-mediated 3' to 5' mRNA decay (*Parker, 2012*). We constructed a set of yeast double mutant strains harboring the *dcp2-N245* allele and deletions of the

*XRN1, SKI2,* or *SKI7* genes. As additional controls, we also constructed yeast double mutants harboring the *dcp2-E153Q-N245* or *dcp2-E198Q-N245* alleles and a deletion of *XRN1.* We selected eleven representative transcripts from the down-regulated group and employed northern blotting to analyze the decay phenotypes of these transcripts in the respective single and double mutant strains. Among the eleven selected transcripts, nine (*GDH1, ARL1, DAL3, YGL117W, RPS9A, SUC2, CPA1, HIS4,* and *SER3*) are typical decapping substrates and two (*HCA1* pre-mRNA and *HSP82* mRNA) are atypical decapping substrates, that is the latter are normally subject to degradation by other decay pathways. As shown in *Figure 2B,* all eleven transcripts manifested decreased levels in *dcp2-N245* cells compared to wild-type cells. Deletion of *XRN1* completely restored the mRNA levels in *dcp2-N245* cells for almost all transcripts. In contrast, elimination of *SKI2* or *SKI7* had no effect on mRNA levels for each of the eleven transcripts in *dcp2-N245* cells. These results validate our RNA-Seq analyses and demonstrate that decreased accumulation of these representative transcripts in *dcp2-N245* cells is indeed caused by accelerated or opportunistic decapping of the mRNAs. This experiment provides direct experimental evidence that Dcp2 is subject to negative regulation through its C-terminal domain and that loss of this negative regulation causes indiscriminate mRNA decapping.

## Decapping activators Pat1, Lsm1, and Dhh1 target specific subsets of yeast transcripts with overlapping substrate specificity

To assess the roles of the general decapping activators Pat1, Lsm1, and Dhh1 in mRNA decay, we generated yeast strains harboring single deletions of the *PAT1, LSM1,* or *DHH1* genes and analyzed the expression profiles of the resulting *pat1Δ, lsm1Δ,* and *dhh1Δ* strains by RNA-Seq. The RNA-Seq libraries from these strains showed good read count distribution (*Figure 3A*) and notable consistency between biological replicates (*Figure 3—figure supplement 1*). Using the same analysis pipeline described above, we identified 940 up-regulated and 685 down-regulated transcripts in *pat1Δ* cells, 955 up-regulated and 681 down-regulated transcripts in *lsm1Δ* cells, and 1098 up-regulated and 788 down-regulated transcripts in *dhh1Δ* cells (*Figure 3B–D*). Because the functions of Pat1, Lsm1, and Dhh1 are required for general mRNA decapping (*Coller and Parker, 2005*; *Fischer and Weis, 2002*; *Coller et al., 2001*; *Tharun et al., 2000*; *Bouveret et al., 2000*), detection of a large number of up-regulated transcripts in *pat1Δ, lsm1Δ,* and *dhh1Δ* cells was expected. The up-regulated transcripts are likely the *bona fide* substrates of these decapping activators, but detection of comparable numbers of down-regulated transcripts in each of these strains was largely unexpected. Much like our observations of the strains with catalytically-deficient Dcp2, deletion of the *PAT1, LSM1,* or *DHH1* genes may have secondary effects on genome-wide mRNA expression.

To explore the functional relationships of Pat1, Lsm1, and Dhh1 in mRNA decay, we compared the up- and down-regulated transcript lists from the *pat1Δ, lsm1Δ,* and *dhh1Δ* strains. As shown in *Figure 3B and C,* transcripts differentially expressed in the *pat1Δ* and *lsm1Δ* strains exhibited highly significant overlap. About 84% of the up-regulated transcripts (864 out of 1031) and 77% of the down-regulated transcripts (583 out of 756) were shared by these two strains. Transcripts differentially expressed in the *dhh1Δ* strain exhibited partial overlap with those in the *pat1Δ* and *lsm1Δ* strains. About 50% of the up-regulated transcripts (542 out of 1098) and 43% of the down-regulated transcripts (342 out of 788) in the *dhh1Δ* strain were shared by the *pat1Δ* or *lsm1Δ* strains. In addition, 482 transcripts were commonly up-regulated and 290 commonly down-regulated in all three strains. Given the substantial overlap of differentially expressed transcripts in the *pat1Δ* and *lsm1Δ* strains, and the physical interactions between Pat1 and the Lsm1-7 complex (*Wu et al., 2014*; *Bouveret et al., 2000*), we tested whether Pat1 and Lsm1 controlled the expression of the same set of transcripts. Utilizing the same differential expression analysis pipeline, we compared the *pat1Δ* and *lsm1Δ* libraries directly. The *pat1Δ* and *lsm1Δ* strains manifested remarkable consistency in their expression profiles over the entire transcriptome, with only four transcripts differentially expressed between the two strains, two of which were caused by the respective gene deletions (*Figure 3E,* leftmost panel, red dots). Direct comparisons were also applied to the *dhh1Δ* and *pat1Δ* or *lsm1Δ* libraries. This analysis revealed significant differences in the expression profiles between the *dhh1Δ* and *pat1Δ* or *lsm1Δ* strains (*Figure 3E,* last two panels). The *dhh1Δ* strain yielded 1332 up-regulated and 874 down-regulated transcripts compared to the *pat1Δ* strain, and 1385 up-regulated and 1037 down-regulated transcripts compared to the *lsm1Δ* strain.

Together, these results indicate that Pat1, Lsm1, and Dhh1 target specific subsets of yeast transcripts with overlapping substrate specificity. Pat1 and Lsm1 appear to function together and target

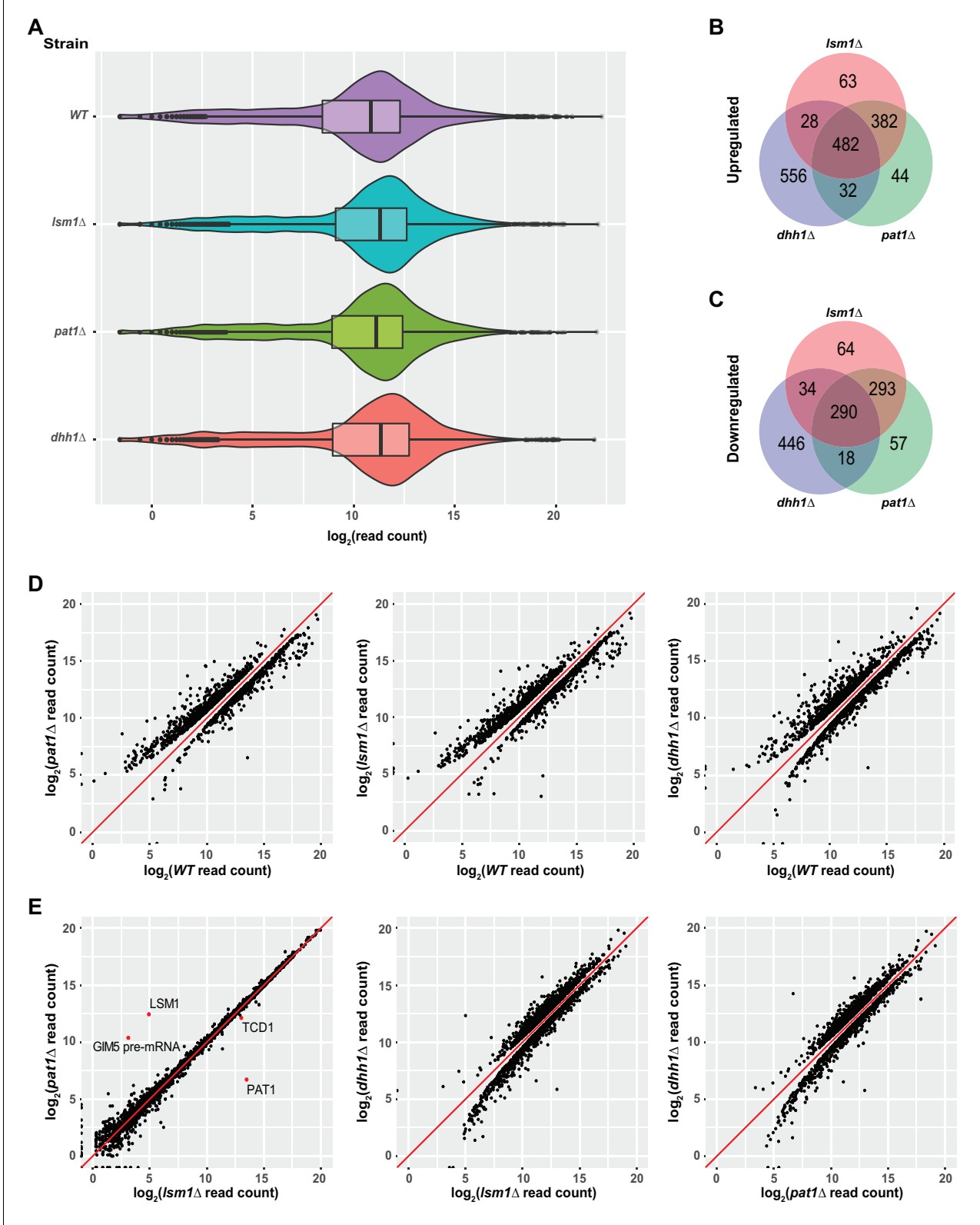

**Figure 3.** Identification of transcripts controlled by Pat1, Lsm1, and Dhh1. (A) Violin and box plots displaying the average and median read count distributions of the RNA-Seq libraries from the *WT*, *pat1Δ*, *lsm1Δ*, and *dhh1Δ* strains in three independent experiments. (B) Venn diagram displaying the relationships between transcripts up-regulated in *pat1Δ*, *lsm1Δ*, and *dhh1Δ* cells. (C) Venn diagram displaying the relationships between transcripts down-regulated in *pat1Δ*, *lsm1Δ*, and *dhh1Δ* cells. (D) Scatterplots comparing the normalized read counts between the *WT* and the *pat1Δ*, *lsm1Δ*, or
*Figure 3 continued on next page*

*Figure 3 continued*

*dhh1Δ* strains for transcripts differentially expressed in each of the mutant strains. Left panel, comparison for the 955 up- and 681 down-regulated transcripts in *pat1Δ* cells; middle panel, comparison for the 940 up- and 685 down-regulated transcripts in *lsm1Δ* cells; and right panel, comparison for the 1098 up- and 788 down-regulated transcripts in *dhh1Δ* cells. (E) Scatterplots comparing the normalized read counts between the *pat1Δ* and *lsm1Δ* strains for all transcripts, or between the *dhh1Δ* strain and the *lsm1Δ* and *pat1Δ* strains for transcripts differentially expressed in these two strains compared to the *dhh1Δ* strain. Left panel, comparison for all transcripts between the *pat1Δ* and *lsm1Δ* strains, four differentially expressed transcripts are indicated by red dots; middle panel, comparison for the 1385 up- and 1037 down-regulated transcripts in the *lsm1Δ* strain with respect to the transcripts of the *dhh1Δ* strain; and right panel, comparison for the 1332 up- and 874 down-regulated transcripts in the *pat1Δ* strain with respect to the transcripts of the *dhh1Δ* strain. For A to E, all analyses were as described in the legend to *Figure 1*.

DOI: https://doi.org/10.7554/eLife.34409.008

The following figure supplements are available for figure 3:

**Figure supplement 1.** RNA-Seq libraries generated from *WT, pat1Δ, lsm1Δ,* and *dhh1Δ* strains exhibit good correlation between three different biological replicates.

DOI: https://doi.org/10.7554/eLife.34409.009

**Figure supplement 2.** Western blotting analysis of Pat1, Lsm1, and Dhh1 levels in different mutant strains.

DOI: https://doi.org/10.7554/eLife.34409.010

the same set of transcripts in yeast cells. Dhh1 appears to have distinct functions from Pat1 and Lsm1 and targets a set of transcripts that only partially overlaps with those regulated by Pat1 and Lsm1. Because Pat1 interacts with both Lsm1 and Dhh1 (*Bouveret et al., 2000*; *Sharif and Conti, 2013*; *Sharif et al., 2013*) we sought to ascertain that the observed overlapping substrate specificities for Pat1, Lsm1, and Dhh1 originated from specific activating functions of these factors, and not from the secondary effects of co-depletion of their interacting partners. Accordingly, we analyzed the levels of Pat1, Lsm1, and Dhh1 in specific gene deletion backgrounds by employing genomic TAP-tagged *PAT1*, *LSM1*, or *DHH1* alleles known to be functional (*Bouveret et al., 2000*; *Fischer and Weis, 2002*). We found that deletion of *PAT1* had little or no effect on the levels of expression of Lsm1 and Dhh1 (*Figure 3—figure supplement 2B and C*). Similarly, deletion of *DHH1* also had no significant effect on the levels of expression of Pat1 and Lsm1 (*Figure 3—figure supplement 2A and B*). Deletion of *LSM1* had no effect on the level expression of Dhh1 (*Figure 3—figure supplement 2C*), but consistent with a previous observation (*Bonnerot et al., 2000*), decreased the level of expression of Pat1 to 35% of its level in wild-type cells (*Figure 3—figure supplement 2A*). The decreased accumulation of Pat1 in the absence of Lsm1 does not contradict our overall interpretation of overlapping mRNA expression patterns between *pat1Δ* and *lsm1Δ* cells, and, in fact, this observation strengthens our conclusion that Pat1 and Lsm1 function together to promote mRNA decapping. Collectively, these results indicate that the observed overlapping substrate specificities of Pat1, Lsm1, and Dhh1 are attributable to their specific activating functions in mRNA decapping.

## Identification of transcripts uniquely and commonly targeted by Pat1, Lsm1, and Dhh1

Based on the well established in vivo functions and in vitro activities of Pat1, Lsm1, and Dhh1 (*Nissan et al., 2010*; *Coller and Parker, 2005*; *Fischer and Weis, 2002*; *Coller et al., 2001*; *Tharun et al., 2000*; *Bouveret et al., 2000*), we considered the up-regulated transcripts in the *pat1Δ*, *lsm1Δ* and *dhh1Δ* strains to be direct substrates of these decapping activators for the most part, and the respective down-regulated transcripts in these strains to arise indirectly as a consequence of general defects in mRNA decapping (see above). To evaluate the reliability of these propositions, we examined the expression patterns of these up- and down-regulated transcripts in *dcp1Δ, dcp2Δ,* and *xrn1Δ* cells as well as in *dcp2-N245, dcp2E-153Q-N245,* and *dcp2E-198Q-N245* cells. Further, to gain insight into the overlapping vs. distinct regulatory activities of Pat1, Lsm1, and Dhh1, we divided the differentially expressed transcripts from *pat1Δ*, *lsm1Δ*, and *dhh1Δ* cells into six distinct subgroups based on their decay phenotypes and examined the distribution of the relative levels of transcripts from these subgroups in each of the mutant strains. The up-regulated transcripts were divided into three non-overlapping subgroups: Up-o-d, up-regulated only in the *dhh1Δ* strain (556 transcripts); Up-o-pl, up-regulated only in the *pat1Δ* and *lsm1Δ* strains (382 transcripts); and Up-a-pld, up-regulated in all three deletion strains (482 transcripts) (*Figure 3B*). Similarly, the down-regulated transcripts were also divided into three non-overlapping subgroups: Down-o-d, down-

regulated only in the *dhh1Δ* strain (446 transcripts); Down-o-pl, down-regulated only in the *pat1Δ* and *lsm1Δ* strains (293 transcripts); and Down-a-pld, down-regulated in all three deletion strains (290 transcripts) (*Figure 3C*).

Transcripts from the six subgroups had distinct expression patterns in *dcp1Δ, dcp2Δ, xrn1Δ, dcp2-N245, dcp2-E153Q-N245,* and *dcp2-E198Q-N245* cells (*Figure 4A–F*). Transcripts from two of the up-regulated subgroups, Up-o-d and Up-a-pld, exhibited similar expression patterns and had significantly increased levels of expression (relative to the WT strain) in all six mutant strains. Transcripts from the third up-regulated subgroup, Up-o-pl, exhibited a slightly different expression pattern and had significantly increased levels in *dcp1Δ, dcp2Δ, dcp2-E153Q-N245,* and *dcp2-E198Q-N245* cells, marginally increased levels in *xrn1Δ* cells, but unaltered levels in *dcp2-N245* cells. These results show that transcripts from the three up-regulated subgroups are all sensitive to the loss of decapping activity, indicating that they are *bona fide* substrates of the decapping enzyme and the general 5' to 3' decay pathway. The marginal effect of *XRN1* deletion on the expression of the transcripts from the Up-o-pl subgroup suggests that, once decapped, a significant fraction of transcripts from this subgroup can also be efficiently degraded by the 3' to 5' decay pathway. This possibility is consistent with experiments demonstrating that deletion of *SKI2* can partially restore the levels of mRNAs downregulated in cells harboring deletions of *PAT1* or *LSM1*. *Figure 4—figure supplement 1* shows that deletion of *SKI2* partially increased the levels of the *GTT2* and *RPP1A* mRNAs in *pat1Δ* and *lsm1Δ* cells, but did not increase the levels of these two mRNAs in *dhh1Δ* cells.

Transcripts from the three down-regulated subgroups exhibited two different expression patterns. Transcripts from the Down-o-d and Down-a-pld subgroups had significantly decreased levels in all six mutant strains. This result shows that transcripts from these two subgroups are sensitive to both partial and complete loss of decapping activity as well as to complete loss of the 5' to 3' exoribonuclease activity. The concordant down-regulation observed for these transcripts in all our mRNA decay mutant strains strongly argues that they are indirectly controlled by the general 5' to 3' decay activities. Transcripts from the down-regulated Down-o-pl subgroup had significantly decreased levels in *dcp1Δ, dcp2Δ, dcp2-E153Q-N245,* and *dcp2-E198Q-N245* cells but increased levels in *xrn1Δ* and *dcp2-N245* cells. The increased expression of these transcripts in response to deletion of *XRN1* and C-terminal truncation of Dcp2 suggests that they are also *bona fide* substrates of the decapping enzyme and thus are likely controlled by Pat1 and Lsm1 directly. Based on our observations in *ski2Δ* cells lacking Pat1 or Lsm1 (see above), the decreased levels of transcripts in *dcp1Δ, dcp2Δ, dcp2-E153Q-N245,* and *dcp2-E198Q-N245* cells suggests that when decapping is completely blocked, these transcripts may also be more efficiently degraded by the 3' to 5' decay pathway.

Collectively, these results indicate that transcripts from all three up-regulated subgroups and one of the down-regulated subgroups (Down-o-pl) are substrates of the decapping enzyme and thus are likely to be direct targets of Pat1, Lsm1, and Dhh1. In contrast, transcripts from the other two down-regulated subgroups (Down-o-d and Down-a-pld) appear to be controlled indirectly by the general 5' to 3' decay activities and thus are not the direct targets of Pat1, Lsm1, and Dhh1. To obtain further support for this conclusion, we analyzed the pattern of the codon protection index for transcripts in each of these six subgroups. This index is a measure of the degree of a transcript's co-translational 5' to 3' decay and is defined as the ratio of sequencing reads in the ribosome protected frame over the average reads of the non-protected frames of the 5' to 3' decay intermediates from a specific transcript (*Pelechano et al., 2015*). Values greater than one are indicators of co-translational 5' to 3' decay. As shown in *Figure 4G*, transcripts from the Up-o-d, Up-o-pl, Up-a-pld, and Down-o-pl subgroups all had median codon protection index values greater than 1. In contrast, transcripts from the Down-o-d and Down-a-pld subgroups both had median codon protection index values less than 1. These data strengthen our conclusion on the separation of differentially expressed transcripts into direct target and non-target categories. Based on their expression patterns, we suggest that transcripts from the Up-o-d subgroup are targeted by Dhh1, transcripts from the Up-o-pl and Down-o-pl subgroups are targeted by Pat1 and Lsm1, and transcripts from the Up-a-pld subgroup are targeted by all three factors.

## Validation of transcripts controlled directly or indirectly by Pat1, Lsm1, and Dhh1

To validate the results from our RNA-Seq analyses and to assess the proposed decay mechanisms for transcripts in different subgroups controlled by Pat1, Lsm1, and Dhh1, we selected 34 transcripts

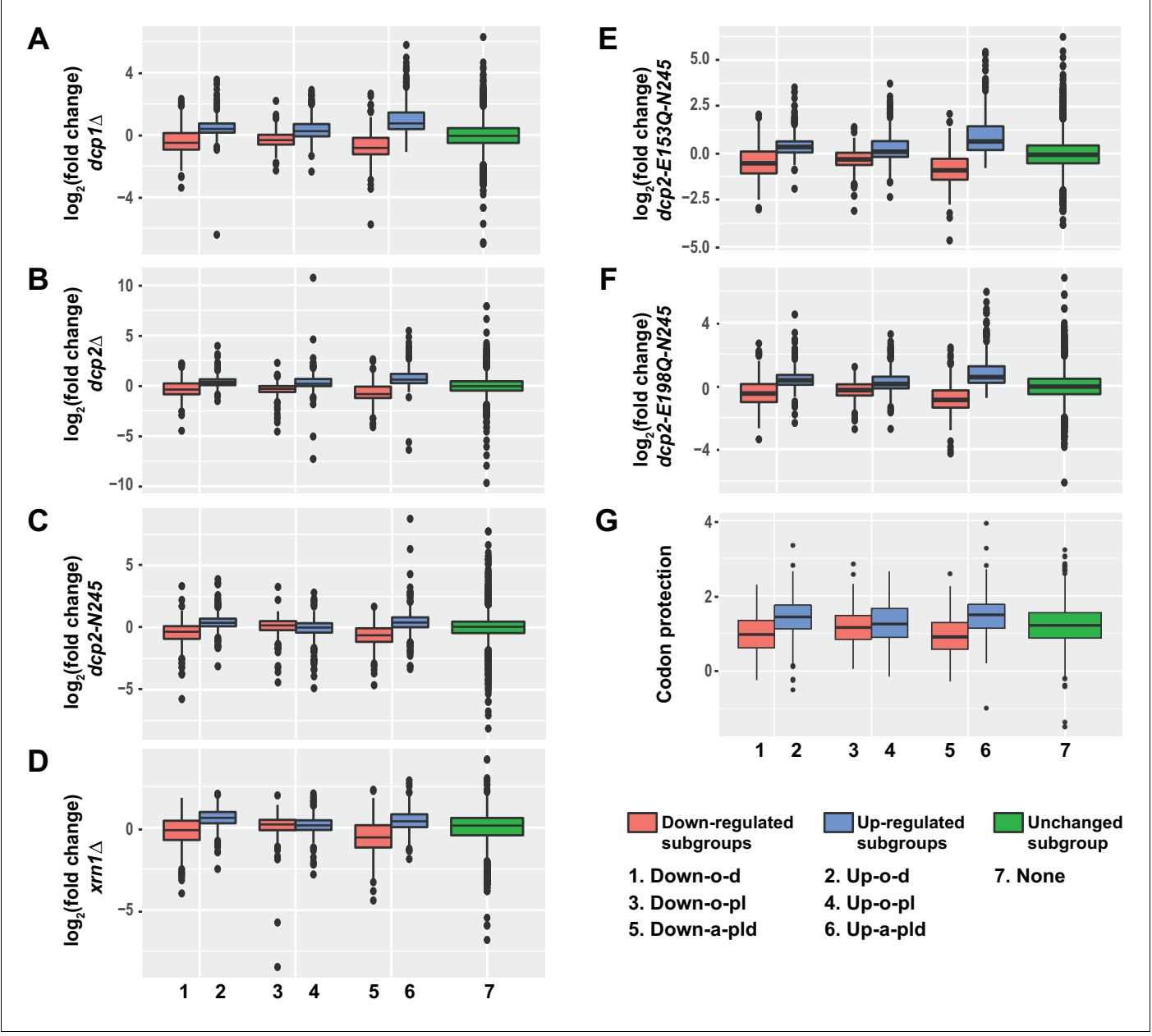

**Figure 4.** Transcripts from different subgroups of mRNAs regulated by Pat1, Lsm1, or Dhh1 have distinct expression patterns in cells deficient in decapping or 5' to 3' exoribonuclease activities and also exhibit distinct extents of co-translational mRNA decay. Transcripts up-regulated in *pat1Δ*, *lsm1Δ*, or *dhh1Δ* strains were divided into three non-overlapping *Up-o-d*, *Up-o-pl*, and *Up-a-pld* subgroups, representing transcripts up-regulated only in *dhh1Δ* cells, only in *pat1Δ* and *lsm1Δ* cells, and in all three deletion strains, respectively. Similarly, transcripts down-regulated in the three deletion strains were also divided into three non-overlapping *Down-o-d*, *Down-o-pl*, and *Down-a-pld* subgroups, representing transcripts down-regulated only in *dhh1Δ* cells, only in *pat1Δ* and *lsm1Δ* cells, and in all three deletion strains, respectively. Transcripts not regulated by Pat1, Lsm1, or Dhh1 were put into the *none* subgroup. Boxplots were used to depict the distributions of both the relative expression levels and the codon protection indices for transcripts in each of these subgroups. In these analyses, the relative expression levels of individual mRNAs in each of the mutant strains were determined by comparisons to the corresponding wild-type strain. The codon protection index of individual mRNAs was based on 5'P seq experiments of wild-type cells under normal growth conditions (*Pelechano et al., 2015*). Log$_2$ transformed data were used to generate all plots except for panel G, and the color codes for the boxplots include: blue for the up-regulated subgroups, red for the down-regulated subgroups, and green for transcripts not regulated by Pat1, Lsm1, or Dhh1. A to F. Boxplots showing the distributions of the relative expression levels for different subgroups in *dcp1Δ* (A), *dcp2Δ* (B), *xrn1Δ* (C), *dcp2-N245* (D), *dcp2-E153Q-N245* (E), and *dcp2-E198Q-N245* (F) cells. (G) Boxplots showing the distributions of the codon protection indices for different subgroups.

DOI: https://doi.org/10.7554/eLife.34409.011

*Figure 4 continued on next page*

*Figure 4 continued*

The following figure supplement is available for figure 4:

**Figure supplement 1.** Inhibition of the 3' to 5' mRNA decay pathway partially restores the levels of transcripts down-regulated in *pat1Δ* and *lsm1Δ* cells, but not in *dhh1Δ* cells.

DOI: https://doi.org/10.7554/eLife.34409.012

(representing mRNAs from each subgroup) and analyzed both their levels and patterns of expression by northern blotting. In this experiment, we analyzed mRNA levels in wild-type, *pat1Δ*, *lsm1Δ*, and *dhh1Δ* strains, but also included *dcp1Δ*, *dcp2Δ*, *xrn1Δ*, *dcp2-N245*, *dcp2-E153Q-N245*, and *dcp2-E198Q-N245* strains to assess each transcript's sensitivity to 5' to 3' decay, and *upf1Δ*, *edc3Δ*, *scd6Δ*, *ski2Δ*, *ski7Δ*, and *ski2Δski7Δ* strains to serve as negative controls. Our northern analyses confirmed the expression patterns for 30 out of 34 selected transcripts. As shown in *Figure 5*, four transcripts (*CIT2*, *SDS23*, *HOS2*, and *PYK2*) from the Up-o-d subgroup all had increased levels only in *dhh1Δ* cells but not in *pat1Δ* and *lsm1Δ* cells; four transcripts (*DIF1*, *AGA1*, *BUR6*, and *LSM3*) from the Up-o-pl subgroup all had increased levels only in *pat1Δ* and *lsm1Δ* cells but not in *dhh1Δ* cells; ten transcripts (*HXT6*, *GPH1*, *HXK1*, *CHA1*, *RTC3*, *NQM1*, *PGM2*, *TMA10*, *GAD1*, and *SPG4*) from the Up-a-pld subgroup all had increased levels in *pat1Δ*, *lsm1Δ*, and *dhh1Δ* cells; and two transcripts (*MUP3* and *GTT2*) from the Down-o-pl subgroup both had decreased levels in *pat1Δ* and *lsm1Δ* cells, but not in *dhh1Δ* cells. Importantly, the twenty transcripts from these four subgroups all had increased levels in *dcp1Δ*, *dcp2Δ*, *dcp2-E153Q-N245*, and *dcp2-E198Q-N245* cells, and nineteen out twenty transcripts (except *GTT2*) also had increased levels in *xrn1Δ* cells. These results support our proposition that transcripts from these four subgroups are all *bona fide* substrates of the decapping enzyme and provide direct evidence that these transcripts are indeed degraded by the general 5' to 3' decay pathway. Also as expected, three transcripts (*RPP1A*, *TMA19*, and *GPD2*) from the Down-a-pld subgroup all had decreased levels in *pat1Δ*, *lsm1Δ*, and *dhh1Δ* cells. Consistent with the idea that these transcripts were affected indirectly as a consequence of a general defect in decapping, all three transcripts also had decreased levels in in *dcp1Δ*, *dcp2Δ*, *dcp2-E153Q-N245*, and *dcp2-E198Q-N245* cells. Interestingly, these three transcripts only had slightly increased levels in *xrn1Δ* cells, suggesting they, too, may be mostly degraded by the 3' to 5' decay pathway. Five transcripts (*YIL164C*, *THI22*, *EST1*, *TRP1-1*, and *ALR2*) from the Down-o-d subgroup had decreased levels only in *dhh1Δ* cells but not in *pat1Δ* and *lsm1Δ* cells (Figure 8D, see below).

The four transcripts that could not be confirmed deviated from expectations for different reasons: one had an extremely low expression level and could not be effectively verified (*SFG1*), one had complex isoforms that are not annotated in the genome releases (*FRE3*), and therefore not used in our statistical procedures, and the other two (*ASC1* pre-mRNA and mRNA) were most likely bioinformatics false positives due to multiple alignment artifacts of sequence reads to the spliced and unspliced isoforms from the same locus.

To ascertain that the increased steady-state levels of the transcripts from the Up-o-d, Up-o-pl, and Up-a-pld subgroups in the respective deletion cells result from changes in mRNA decay rates, we constructed a set of yeast *rpb1-1* strains containing deletions of *PAT1*, *LSM1*, or *DHH1* and determined half-lives for fourteen transcripts that had been validated by northern blotting (above). As shown in *Figure 6A and B*, two representative transcripts (*SDS23* and *PYK2*) from the Up-o-d subgroup both had increased half-lives in *dhh1Δ* cells, that is from 8.0 and 12.6 min respectively in wild-type cells to 21.0 min in each case in *dhh1Δ* cells. Similarly, compared to wild-type cells, three representative transcripts (*DIF1*, *BUR6*, and *LSM3*) from the Up-o-pl subgroup all had increased half-lives in *pat1Δ* and *lsm1Δ* cells. Finally, three representative transcripts (*CHA*, *HXT6*, *GPH1*, and *HXK1*) from the Up-a-pld subgroup all had increased half-lives in *pat1Δ*, *lsm1Δ*, and *dhh1Δ* cells. We also analyzed the decay rates for six additional transcripts (*GPH1*, *RTC3*, *NQM1*, *PGM2*, *TMA10*, and *GAD1*) from the Up-a-pld subgroup, but were unable to generate reliable half-lives for these transcripts. These six transcripts are all encoded by annotated stress responsive genes and their transcription could not be inhibited by either thermal inactivation of *rpb1-1* or by treating cells with thiolutin (data not shown). These results indicate that the increased steady-state accumulation of transcripts from the three subgroups in *pat1Δ*, *lsm1Δ*, and *dhh1Δ* cells are most likely direct consequences of the loss of Pat1, Lsm1, and Dhh1 functions in mRNA decapping and suggest that at least

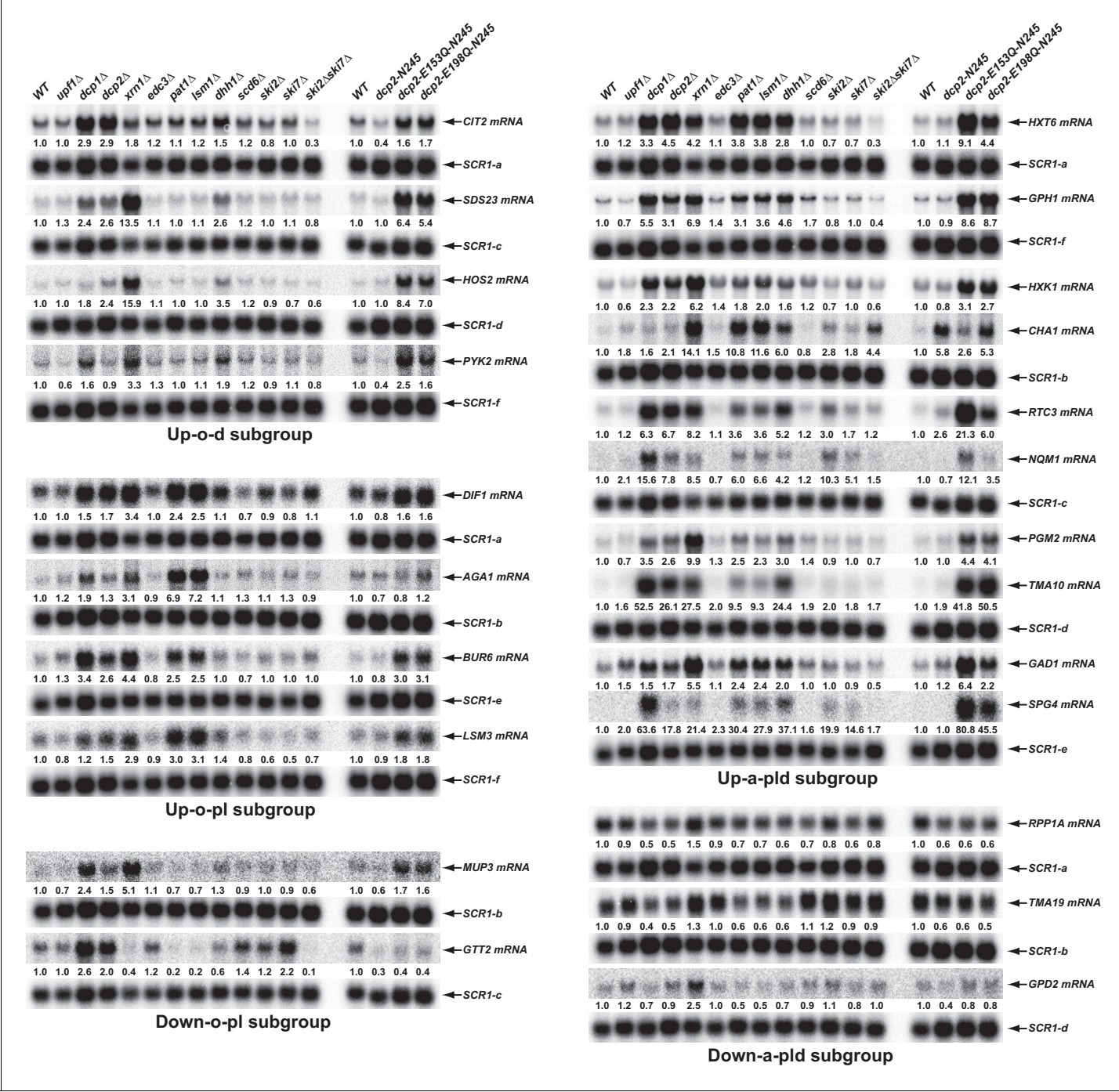

**Figure 5.** Validation of representative transcripts regulated by Pat1, Lsm1, or Dhh1. Representative transcripts from five of the subgroups (Up-o-d, Up-o-pl, Up-a-pld, Down-o-pl, and Down-a-pld) described in *Figure 4* were selected and their steady-state levels in the indicated strains were analyzed. Since individual blots were reprobed multiple times for different transcripts some *SCR1* blots served as controls for different transcripts. The shared *SCR1* blots are indicated by lower case letters a, b, c, d, e, and f, respectively.
DOI: https://doi.org/10.7554/eLife.34409.013

a significant fraction of transcripts from these three subgroups are direct targets of Pat1, Lsm1, and Dhh1. In addition, our observation that Pat1, Lsm1, and Dhh1 function in repressing the expression of transcripts from stress responsive genes suggests that the regulatory activities of these three factors may also play an important role in cellular responses to environmental stresses.

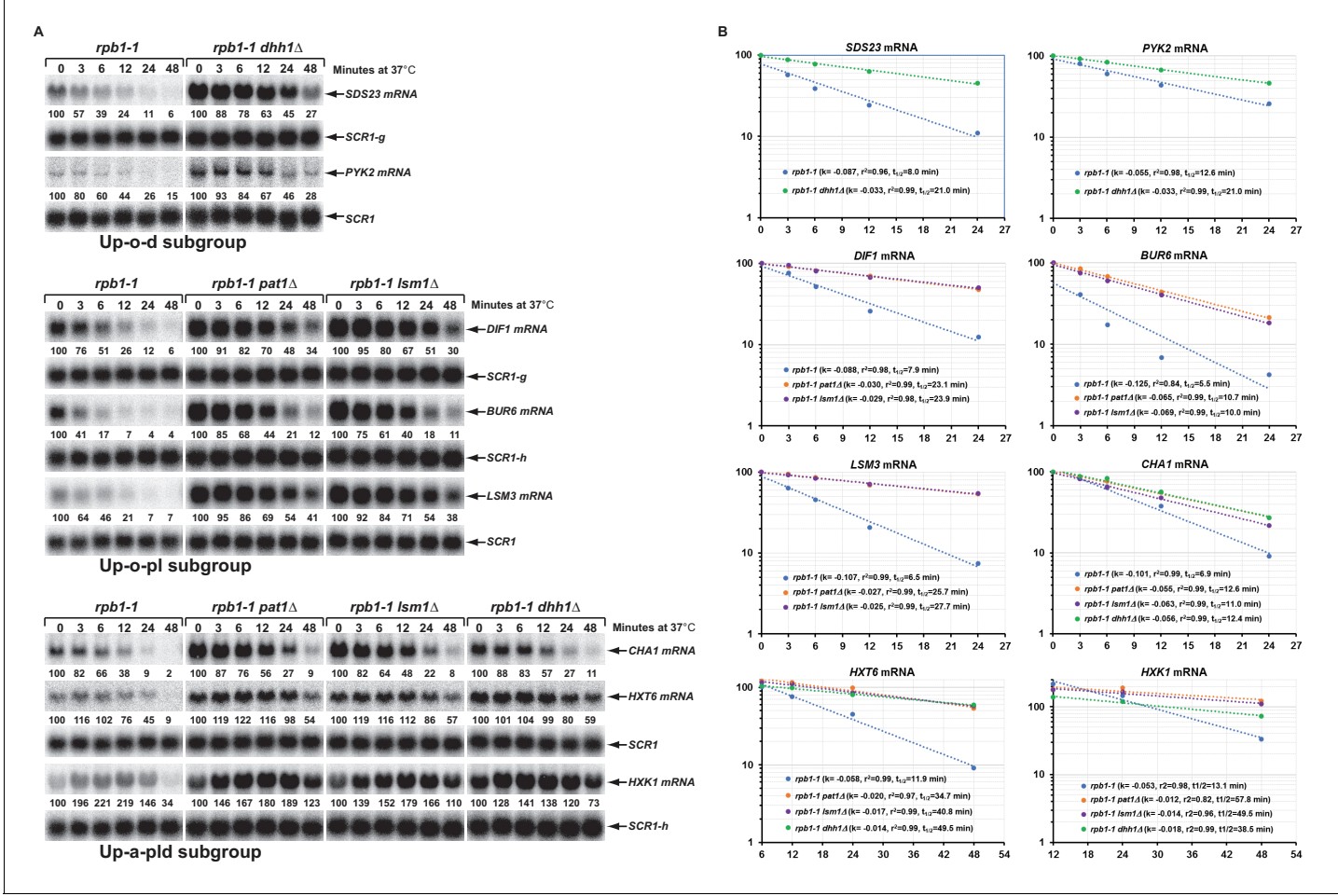

**Figure 6.** Decay rates of representative transcripts regulated by Pat1, Lsm1, or Dhh1. Representative transcripts from the upregulated subgroups described in *Figure 4* (Up-o-d, Up-o-pl, and Up-a-pld) were selected and their decay rates were determined by northern blot quantitation of the fraction of mRNA remaining after a temperature-shift in *rpb1-1*, *rpb1-1/pat1Δ*, *rpb1-1/lsm1Δ*, and *rpb1-1/dhh1Δ* cells. The quantitative data (% of mRNA remaining) at different time points after the temperature shift relative to time point zero for each transcript shown in panel A is plotted in panel B. For each graph in panel B, the Y-axis represents the percentage of mRNA remaining and the X-axis represents time in minutes. In each case, the data were fitted to a single exponential equation and the slope (k) and the squared regression co-efficiency ($r^2$) of the regression line were determined. Half-lives of each transcript in different strains were calculated using the formula $T_{1/2}=\ln2/k$. Data from the first five time points were used for graphing *SDS23*, *PYK2*, *DIF1*, *BUR6*, and *CHA1* mRNA decay rates. However, data from the last four and last three time points were used for graphing *HXT6* and *HXK1* mRNA decay rates, respectively, as the latter two mRNAs exhibited delayed transcriptional inhibition during the temperature shift. As noted in the legend to *Figure 5*, individual blots were reprobed for different transcripts. The shared *SCR1* blots are indicated by lower case letters g and h.
DOI: https://doi.org/10.7554/eLife.34409.014

## Transcripts targeted by Pat1, Lsm1, and Dhh1 are all translated inefficiently

Given the intimate linkage of mRNA translation and decay (*Mishima and Tomari, 2016*; *Presnyak et al., 2015*; *Roy and Jacobson, 2013*), and to gain insight into the roles of Pat1, Lsm1, and Dhh1 in decapping regulation, we sought to identify any unique properties associated with the translation of transcripts controlled by these three factors. To this end, we analyzed the pattern and distribution of the average codon optimality score, the average ribosome density, and the estimated protein abundance for transcripts from the six subgroups of differentially expressed mRNAs in *pat1Δ*, *lsm1Δ*, and *dhh1Δ* cells. In this analysis, the average codon optimality score of individual transcripts was based on the scores defined by Pechmann and Frydman (*Pechmann and Frydman, 2013*) and these scores are in fact the normalized tRNA adaptation index in which mRNA abundances are used to correct for the number of codons vs. number of tRNA genes. Ribosomal densities

were derived from published ribosome profiling and RNA-Seq data from wild-type yeast cells grown under standard conditions (*Young et al., 2015*). Protein abundance levels were obtained from the curated PaxDb (Protein Abundances Across Organisms) database (*Wang et al., 2012*) and comprise the scaled aggregated estimates over several proteomic data sets.

As shown in *Figure 7A–C*, transcripts from the three up-regulated Up-o-d, Up-o-pl, and Up-a-pld subgroups exhibited similar and consistent data patterns: they all had relatively low average codon

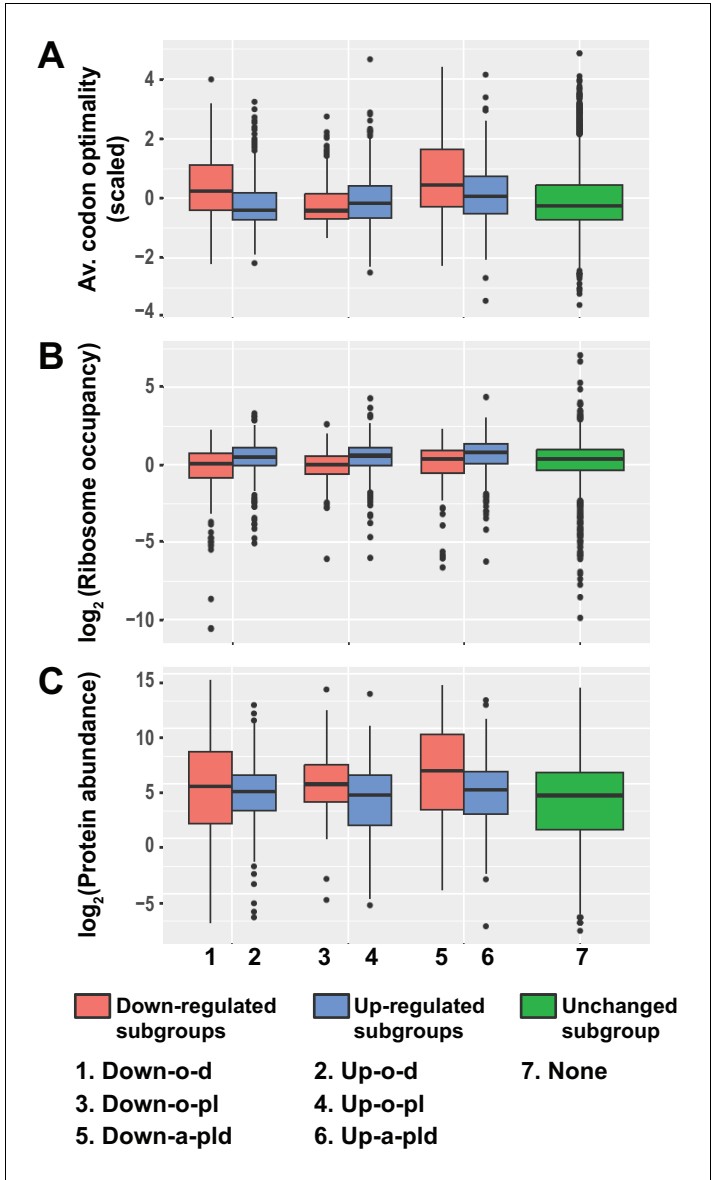

**Figure 7.** Transcripts from different subgroups of mRNAs regulated by Pat1, Lsm1, and Dhh1 have distinct translational properties. Boxplots were used to examine the distributions of average codon optimality scores, ribosome occupancies, and scaled protein abundances for transcripts from each of the six regulation subgroups described in *Figure 4*. In this analysis, codon optimality scores are based on the normalized tRNA adaptation index (*Pechmann and Frydman, 2013*), ribosome occupancies are based on ribosome footprint profiling data of wild-type cells under normal growth conditions (*Young et al., 2015*), and protein abundance scores are based on curated data in a database (*Wang et al., 2015*; *Wang et al., 2012*). A to C. Boxplots showing the distributions of scaled average codon optimality scores (**A**) and Log$_2$ transformed data of ribosome occupancies (**B**) or scaled protein abundances (**C**). Boxplots are color coded as described in the legend to *Figure 4*.
DOI: https://doi.org/10.7554/eLife.34409.015

optimality scores, high ribosome densities, and low protein levels. These observations indicate that transcripts from these three subgroups are all translated inefficiently, most likely because of less efficient translation elongation. Transcripts from the three down-regulated subgroups exhibited two different data patterns. Transcripts from the Down-o-d and Down-a-pld subgroups exhibited one consistent data pattern: they all had relatively high average codon optimality scores, low ribosome densities, and high protein levels. These observations suggest that transcripts from these two subgroups are translated efficiently, a characteristic probably reflecting highly efficient translation elongation. Transcripts from the Down-o-pl subgroup had a distinct data pattern: they had relatively low average codon optimality scores and low ribosome densities, but relatively high protein levels. These observations suggest that transcripts from this subgroup may be inefficiently translated, but have relatively long mRNA half-lives. Together, these results indicate that transcripts targeted directly by Pat1, Lsm1, and Dhh1 are all translated less efficiently. In contrast, transcripts controlled indirectly by these three factors appear to be translated more efficiently.

Notably, although the Dhh1 function in mRNA decay was recently suggested to be linked to codon optimality (*Radhakrishnan et al., 2016*), our analyses revealed that the average codon optimality score of individual transcripts was not a reliable predictor of the Dhh1 requirement for their decay. Transcripts targeted by Dhh1 (from the Up-o-d and Up-a-pld subgroups) had a low but broad range of average codon optimality scores (*Figure 7A*). Interestingly, in that range, there were also thousands of transcripts that were not targeted by Dhh1, including transcripts uniquely targeted by Pat1 and Lsm1 (from the Up-o-pl and Down-o-pl subgroups), as well as transcripts targeted by none of these three factors. This raises the possibility that additional decay factors may be responsible for targeting these transcripts.

## Pat1, Lsm1, and Dhh1 have non-overlapping functions with NMD factors in mRNA decapping regulation

To further define the roles of Pat1, Lsm1, and Dhh1 in mRNA decapping regulation, we examined the functional relationships between these three factors and the NMD factors Upf1, Upf2, and Upf3. We compared the transcripts targeted by Pat1, Lsm1, and Dhh1 to those targeted by the three Upfs (*Celik et al., 2017*). As shown in *Figure 8A*, transcripts from the Up-o-d, Up-o-pl, and Up-a-pld subgroups all had only minimal and insignificant overlap with NMD substrates. In addition, as revealed by a two-dimensional hierarchical clustering analysis of differentially expressed transcripts, *pat1Δ*, *lsm1Δ*, and *dhh1Δ* cells also had profiles distinct from those of *upf1Δ*, *upf2Δ*, and *upf3Δ* cells (*Figure 8B*). These results indicate that the general decapping activators Pat1, Lsm1, and Dhh1 have roles that are distinct from and non-overlapping with those of the NMD factors in mRNA decapping regulation.

## Deletion of *DHH1* promotes the degradation of a fraction of NMD substrates

We recently demonstrated that yeast decapping activators form distinct complexes with the decapping enzyme in vivo (*He and Jacobson, 2015a*), suggesting that different decapping activators may compete with each other for binding to the decapping enzyme. One implication of this notion is that in addition to providing targeting specificity for the decapping enzyme, decapping activators can also control each other's activities indirectly by limiting or promoting the free pool of available decapping enzyme. A testable prediction of this dynamic mRNA decapping regulation is that, in addition to stabilizing its targeted transcripts, deletion of a specific activator may also promote the degradation of substrates of the alternative mRNA decay pathways. To test this hypothesis, we examined whether the down-regulated subgroups from the differentially expressed transcripts in *pat1Δ*, *lsm1Δ*, and *dhh1Δ* cells may contain NMD substrates. This analysis revealed that the three subgroups exhibited significantly different enrichment patterns for NMD substrates (*Figure 8C*). The Down-o-d subgroup (transcripts down-regulated only in *dhh1Δ* cells) was enriched for NMD substrates (Fisher's exact test, $p = 7.3 \times 10^{-15}$). In contrast, the Down-o-pl (transcripts down-regulated only in in *pat1Δ* and *lsm1Δ* cells) was depleted of NMD substrates (Fisher's exact test, $p = 6.2 \times 10^{-6}$). Finally, the Down-a-pld subgroup (transcripts down-regulated in all three deletion strains) showed neither enrichment for nor depletion of NMD substrates (Fisher's exact test $p = 0.12$). These results provide additional evidence that transcripts from the Down-o-pl subgroup

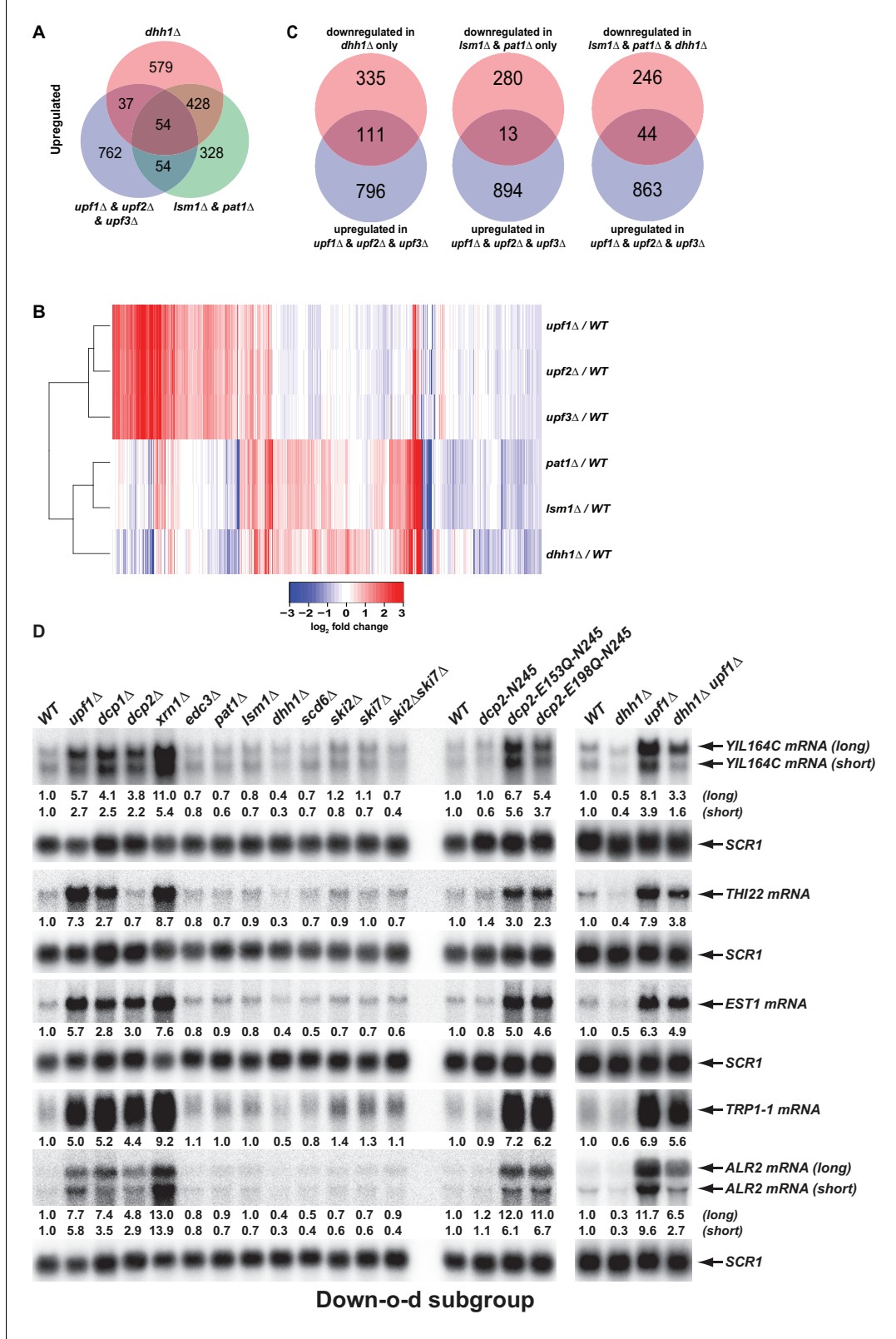

**Figure 8.** Decapping activators have distinct targeting specificities and display dynamic regulation. (A) Venn diagram depicting minimal significant overlaps between transcripts targeted by the Upf factors and those targeted by Dhh1 or Pat1 and Lsm1. (B) Two-dimensional clustering analysis of differentially expressed transcripts showing distinct expression patterns of yeast cells harboring deletions of the *UPF1*, *UPF2*, *UPF3*, *PAT1*, *LSM1*, or *DHH1* genes. The relative levels of individual mRNAs in the deletion strains were determined by comparisons to the corresponding wild-type strain.
*Figure 8 continued on next page*

*Figure 8 continued*

Log₂ transformed ratios were used for clustering analyses. The data for the NMD factors were from our previous study (*Celik et al., 2017*). Color coding used to represent fold change in expression employs red to indicate increases in levels and blue to indicate decreases in levels, with intermediate changes scaled to lighter versions of each color. (C) Venn diagrams depicting the enrichment of NMD-targeted transcripts in the Down-o-d subgroup, but not in the Down-a-pld and Down-o-pl subgroups of mRNAs indirectly controlled by Pat1, Lsm1, and Dhh1. (D) Northen blotting analysis of representative transcripts from the Down-o-d subgroup of mRNAs that are targeted by NMD. Five transcripts were selected, and northern blotting and transcript quantification were as described in the legend to *Figure 2B*.
DOI: https://doi.org/10.7554/eLife.34409.016

are direct targets of Pat1 and Lsm1, and show that the Down-o-d subgroup contains a fraction of NMD-targeted transcripts. To validate the latter observation, we selected five representative NMD substrates that were also down-regulated in *dhh1Δ* cells and analyzed their expression levels and patterns in a set of yeast strains described above. As expected, northern analyses showed that all five transcripts (*YIL164C*, *THI22*, *EST1*, *TRP1-1*, and *ALR2*) had decreased levels in *dhh1Δ* cells but increased levels in *upf1Δ*, *dcp1Δ*, *dcp2Δ*, and *xrn1Δ* cells (*Figure 8D*, left panel). The decreased accumulation for these five transcripts in *dhh1Δ* cells largely resulted from degradation by NMD as elimination of *UPF1* from *dhh1Δ* cells caused substantial increases (ranging from 4.0- to 21.6-fold) in the expression levels of each of these five transcripts (*Figure 8D*, right panel). Interestingly, we also observed that *dhh1Δ upf1Δ* cells consistently accumulated lower levels (ranging from 28% to 81%) than *upf1Δ* cells for each of these five transcripts (*Figure 8D*, right panel), suggesting that deletion of *DHH1* can also promote NMD-independent degradation of these transcripts. Together, these results indicate that deletion of *DHH1* can promote the degradation of a subset of NMD substrates by both NMD-dependent and NMD-independent mechanisms, thus arguing that decapping activators can indeed exert indirect control of each other's activities in mRNA decapping. This may indicate that the decapping enzyme is limiting under normal growth conditions.

## Discussion

### Dcp2 C-terminal domain imparts critical in vivo regulatory activities in mRNA decapping

The yeast Dcp2 decapping enzyme subunit has a modular structure encompassing a conserved 245-amino acid N-terminal Nudix catalytic domain and a 725-amino acid C-terminal extension. While the catalytic function of the N-terminal domain in cap removal is well established (*Floor et al., 2010*; *She et al., 2008*; *Deshmukh et al., 2008*; *She et al., 2006*), the decapping role of the large Dcp2 C-terminal domain remains to be clarified. Here, we provide genetic evidence that the Dcp2 C-terminal domain imparts important regulatory activities to the decapping enzyme, thus playing a critical role in regulating mRNA decapping in vivo. Elimination of the C-terminal domain altered the expression of more than a quarter of yeast's protein-coding genes and led to both up- and down-regulation of specific transcripts (*Figure 1B,C*). A key observation supporting a predominantly regulatory role for the C-terminal domain is that transcripts differentially expressed in cells lacking the Dcp2 C-terminal domain only exhibited limited correlation with those differentially expressed in cells whose decapping activity was either severely comprised or essentially absent (*Figure 2A*).

Our recent experiments revealed that the Dcp2 C-terminal domain harbors both negative and positive regulatory elements (*He and Jacobson, 2015a*), leading us to propose that the decapping enzyme is subject to both negative and positive regulation. Recent biochemical data also supports this model (*Paquette et al., 2018*). Our expression profiling of yeast cells lacking the Dcp2 C-terminal domain provides direct experimental evidence for both aspects of this hypothesis. Negative regulation of the decapping enzyme is supported by the observations that deletion of the Dcp2 C-terminal domain led to decreases in the abundance of hundreds of specific transcripts, and that this down-regulation was dependent on maintenance of Dcp2's catalytic activity (*Figures 1C* and *2B*). Importantly, the vast majority of these down-regulated transcripts were not normal decapping substrates (*Figure 1—figure supplement 3*). These results indicate that deletion of the Dcp2 C-terminal domain eliminates an inhibitory function of the domain and leads to uncontrolled and accelerated mRNA decapping by a constitutively activated and opportunistic Dcp2. Evidence for positive

regulation of the decapping enzyme is provided by the observation that elimination of the Dcp2 C-terminal domain also caused up-regulation of hundreds of specific transcripts. As this group of transcripts also exhibited concordant up-regulation in decapping-deficient cells (*Figure 1B*), it is likely that the observed up-regulation originates from a deficiency in mRNA decapping caused by loss of a positive regulatory function. Collectively, these observations indicate that the C-terminal domain of Dcp2 encodes important regulatory activities and that loss of these regulatory activities can have direct consequences on decapping of hundreds of specific mRNAs.

Over the past decade mechanistic investigations of mRNA decapping regulation have largely been focused on the 245-amino acid N-terminal domain of Dcp2, with essentially all biochemical and structural studies using this C-terminally truncated fragment (*Mugridge et al., 2016*; *Borja et al., 2011*; *Floor et al., 2010*; *She et al., 2008*; *Deshmukh et al., 2008*; *She et al., 2006*; *Wurm et al., 2017*; *Wurm et al., 2016*; *Charenton et al., 2016*; *Mugridge et al., 2018*). This Dcp2 fragment binds to Dcp1, but lacks the binding sites for most decapping activators, including Pat1, Edc3, and Upf1 (*He and Jacobson, 2015a*). Our genetic experiments here reveal that this N-terminal fragment of Dcp2 encodes a constitutively active decapping enzyme in vivo that can target a variety of mRNAs including those that normally use or do not use decapping-dependent mechanisms in their degradation (*Figure 1B–C*). Accordingly, current models of mRNA decapping regulation based on the Dcp2 N-terminal domain may be informative with respect to the catalytic step of decapping, but most likely do not reflect complex aspects of mRNA decapping regulation in vivo such as substrate selection and decapping enzyme activation.

## Pat1, Lsm1, and Dhh1 target subsets of yeast transcripts with overlapping substrate specificity

Pat1, Lsm1, and Dhh1 have long been considered as general mRNA decapping activators and their functions are usually thought to be required for decapping of most wild-type mRNAs (*Parker, 2012*; *Coller and Parker, 2004*; *Fischer and Weis, 2002*; *Coller et al., 2001*; *Bouveret et al., 2000*; *Tharun and Parker, 1999*). Contrary to this expectation, our expression profiling experiments revealed that Pat1, Lsm1, and Dhh1 are only required for decapping of a subset of transcripts in yeast cells and suggested that these factors have highly specific functions in controlling mRNA decapping (*Figure 3B*). Consistent with strong in vivo physical interaction and shared in vitro RNA binding properties (*Wu et al., 2014*; *Sharif and Conti, 2013*; *Bouveret et al., 2000*; *Chowdhury et al., 2007*), our results indicate that Pat1 and Lsm1 function together (probably as a Pat1-Lsm1-7 complex) to target the same set of transcripts (*Figure 3B–E*). Dhh1 targets a different set of transcripts that only partially overlaps with those targeted by both Pat1 and Lsm1 (*Figure 3B*).

The partial overlap between transcripts commonly targeted by Pat1 and Lsm1 and those targeted by Dhh1 strongly indicates that these three decapping activators have distinct functions in mRNA decapping regulation and that decapping of individual mRNAs likely has different functional requirements for Pat1, Lsm1, and Dhh1. For example, we identified transcripts regulated by Dhh1 but not by Pat1 and Lsm1 (the Up-o-d subgroup), transcripts regulated by Pat1 and Lsm1 but not by Dhh1 (the Up-o-pl and Down-o-pl subgroups), and transcripts regulated by all three factors (the Up-a-pld subgroup). Since the degradation of transcripts controlled by Pat1, Lsm1, and Dhh1 individually or in combination is dependent on the Dcp1-Dcp2 decapping enzyme and the Xrn1 5' to 3' exoribonuclease (*Figures 4* and *5*) the transcripts in these three groups are most likely *bona fide* decapping substrates. Accordingly, our observation that decapping of individual mRNAs can have different requirements for Pat1, Lsm1, and Dhh1 suggests that mRNA decapping is a multi-step process and that Pat1, Lsm1, and Dhh1 are likely to function at different steps of the decapping pathway. Given the genetic and physical interactions between Dhh1 and the Not1-Ccr4 deadenylase complex (*Chen et al., 2014*; *Rouya et al., 2014*; *Maillet and Collart, 2002*; *Hata et al., 1998*; *Ozgur et al., 2015*; *Mathys et al., 2014*), and the physical interaction between Pat1 and Dcp2 (*Charenton et al., 2017*; *He and Jacobson, 2015a*), one possibility is that Dhh1 promotes deadenylation and that Pat1 and Lsm1 recruit the decapping enzyme. In addition, since Pat1 and Lsm1, but not Dhh1, also interact with Xrn1 (*Charenton et al., 2017*), the former two factors may also control a step after decapping, that is 5' to 3' exonucleolytic digestion.

Transcripts targeted by Pat1, Lsm1, and Dhh1 all appear to be translated inefficiently as they have relatively low average codon optimality scores, high ribosomal occupancy, and low protein production (*Figure 7*). These observations indicate that the functions of Pat1, Lsm1, and Dhh1 in

regulating decapping are probably linked to mRNA translation, a conclusion consistent with a recent study linking Dhh1 function in mRNA decay to translation elongation through codon optimality (*Radhakrishnan et al., 2016*). However, our results indicate that average codon optimality scores of individual mRNAs do not correlate well with a Dhh1 requirement for their decay (*Figure 7*). Hence, the identity and distribution of non-optimal codons in an mRNA may influence the targeting specificity by Pat1, Lsm1, and Dhh1.

Over past two decades models for the general functions of Pat1, Lsm1, and Dhh1 in yeast mRNA decapping were largely generated by assessing the fate of the transcripts derived from two key reporter gene constructs (*Fischer and Weis, 2002*; *Coller et al., 2001*; *Hatfield et al., 1996*; *Decker and Parker, 1993*). One reporter codes for the unstable *MFA2* mRNA and the other codes for the stable *PGK1* mRNA. Both of these transcripts are in the datasets presented here and our results show that the *MFA2* mRNA is regulated by Pat1 and Lsm1, but not by Dhh1, and that the *PGK1* mRNA is not regulated by any of the three factors. These observations highlight potential drawbacks to the use of reporter gene assays and lead to uncertainty about existing models. Most importantly, since both reporter mRNAs are not regulated by Dhh1 it becomes difficult to justify models in which Dhh1 has a direct role in decapping of these transcripts (*Coller and Parker, 2005*; *Fischer and Weis, 2002*; *Coller et al., 2001*).

## Pat1, Lsm1, and Dhh1 also have indirect roles in controlling genome-wide mRNA expression

Our expression profiling experiments revealed that, in addition to targeting specific mRNAs for decapping, Pat1, Lsm1, and Dhh1 also have indirect roles in controlling mRNA expression in yeast and that eliminating the functions of any of these three factors can have severe consequences for global mRNA accumulation. Unexpectedly, we found that deletions of *PAT1*, *LSM1*, and *DHH1* also resulted in down-regulation of hundreds of specific transcripts (*Figure 3C*). In contrast to the up-regulated transcripts, the vast majority of the down-regulated transcripts are not direct targets of Pat1, Lsm1, or Dhh1 and their down-regulation is thus likely to result from an indirect consequence of losing the primary functions of these factors in mRNA decapping. Two subgroups of transcripts that are indirectly controlled by the activities of Pat1, Lsm1, and Dhh1 were identified. One subgroup (Down-o-d) includes transcripts that were down-regulated only in *dhh1Δ* cells and the other subgroup (Down-a-pld) includes transcripts that were down-regulated in all three deletion strains. The transcripts from these two subgroups exhibited concordant down-regulation in yeast cells partially compromised in, or completely lacking, decapping activity (*Figure 4A–F*). Collectively, these observations indicate that transcripts from the Down-o-d and Down-a-pld subgroups are sensitive to loss of both the regulatory and catalytic activities of mRNA decapping and argue that they are indirectly controlled by the status of general decapping activity in yeast cells.

Unlike the transcripts targeted directly by Pat1, Lsm1, and Dhh1, transcripts controlled indirectly by these factors are translated efficiently. Transcripts from the Down-o-d and Down-a-pld subgroups generally have higher average codon optimality scores, lower ribosomal occupancy, and higher protein production (*Figure 7*). These observations suggest that the susceptibility of these transcripts to the loss of Pat1, Lsm1, and Dhh1 functions is likely to be dictated by their unique properties in translation. The Down-o-d subgroup contains a small set of NMD substrates (*Figure 8C*) and the decreased accumulation of these NMD-regulated transcripts in *dhh1Δ* cells largely results from more efficient decapping by an NMD-dependent mechanism (*Figure 8D*, right panel). Because Dhh1 forms several distinct complexes with the decapping enzyme (*Sharif et al., 2013*; *Fromm et al., 2012*; *Tritschler et al., 2009*), more efficient decapping of NMD substrates in the absence of Dhh1 is likely caused by increases in the free pool of the decapping enzyme available for NMD as a consequence of *DHH1* deletion. The majority of transcripts controlled indirectly by Pat1, Lsm1, and Dhh1 are not typical decapping substrates (*Figures 4* and *5*). The decreased accumulation of these transcripts in the absence of Pat1, Lsm1, and Dhh1 probably results from more efficient 3' to 5' degradation. One possibility is that these transcripts are normally protected by an unknown factor at their 3'-ends. Inactivation of Pat1, Lsm1, and Dhh1 leads to the stabilization of a significant number of transcripts. The stabilized transcripts might sequester the unknown factor from their normal binding substrates and make the latter susceptible to 3' to 5' decay. An interesting implication of these observations is that deletion of the genes encoding regulators of other steps in the gene expression pathway may lead to similar indirect and opportunistic effects.

## Reassessing the major functions of decapping activators

Current models of mRNA decapping propose two major temporally separated functions for decapping activators, an initial repression of mRNA translation followed by stimulation of the activity of the decapping enzyme (*Parker, 2012*; *Nissan et al., 2010*; *Coller and Parker, 2005*). Thus, for example, Dhh1 is thought to function principally in repressing translation, Edc3 is thought to activate the decapping enzyme, and Pat1 is thought to possess both activities (*Nissan et al., 2010*; *Coller and Parker, 2005*). It is also generally believed that decapping of individual mRNAs requires the functions of multiple decapping activators (*Nissan et al., 2010*; *Coller and Parker, 2005*). Our results from in vivo expression profiling experiments presented here, and genetic analyses published earlier (*He and Jacobson, 2015a*), challenge these views with data indicating that: a) the main function of decapping activators is to provide substrate specificity, that is to target the decapping enzyme to specific mRNAs; b) individual decapping activators target highly specific subsets of yeast transcripts and generally do not have overlapping regulatory activities (*Figure 8A*); and c) the decapping enzyme is subject to negative regulation and its activation is most likely coupled to substrate recognition. These principles are not dependent on translational repression, and accumulating experimental evidence indicates that prior translational repression may not be required for decapping to occur. Decapping of individual mRNAs occurs while they are still engaged in translation (*Hu et al., 2010*; *Hu et al., 2009*) and this co-translational decay appears to be widespread, both in genome-wide analyses (*Pelechano et al., 2015*) and in our experiments evaluating the transcripts targeted by Dhh1 (*Figure 4G*). Thus, decapping activators may not have primary roles in regulating mRNA translation, but instead may function by monitoring mRNA translation initiation, elongation, or termination to target unique features of individual mRNAs.

## Materials and methods

**Key resources table**

| Reagent type (species) or resource | Designation | Source or reference | Identifiers | Additional information |
|---|---|---|---|---|
| Chemical compound, drug | [α-$^{32}$P]-dCTP | Perkin Elmer | Blu513Z | |
| Chemical compound, drug | Herring Sperm DNA | Promega | D1815 | |
| Peptide, recombinant protein | Taq DNA polymerase | Roche | 04-728-874-001 | |
| Peptide, recombinant protein | Baseline-ZERO DNase | Epicentre | DB0711K | |
| Peptide, recombinant protein | SuperScript II Reverse Transcriptase | Invitrogen | 18064–022 | |
| Strain, strain background (W303 or BY4741) | Yeast strains used in this study | This paper | *Supplementary File 1* | Contains all yeast strains obtained or constructed in this study |
| Recombinant DNA reagent | Plasmids used in this study | This paper | *Supplementary File 2* | Contain all plasmids constructed in this study |
| Sequence-based reagent | Oligonucleotides used in this study | This paper | *Supplementary File 3* | Contains all oligonucleotides used in this study |
| Antibody | Mouse anti-HA monoclonal antibody | Sigma | H3663 | one to 4000 |

*Continued on next page*

*Continued*

| Reagent type (species) or resource | Designation | Source or reference | Identifiers | Additional information |
|---|---|---|---|---|
| Antibody | Rabbit anti-TAP Tag polyclonal antibody | Thermo Fisher | CAB1001 | one to 1000 |
| Antibody | Mouse anti-Pgk1 monoclonal antibody | Invitrogen | 459250 | one to 8000 |
| Commercial assay or kit | Ribo-Zero Gold rRNA Removal Kit (Yeast) | Illumina | MRZY1306 | |
| Commercial assay or kit | TruSeq Stranded mRNA LT Sample Prep Kit | Illumina | RS-122–2101 | |
| Commercial assay or kit | Agencourt RNA Clean XP Kit | Beckman-Coulter Genomics | A63987 | |
| Commercial assay or kit | Random Primed DNA labeling Kit | Roche | 11-004-760-001 | |
| Software, algorithm | RSEM | *Li and Dewey, 2011* | http://deweylab.biostat.wisc.edu/rsem | |
| Software, algorithm | DESeq | *Anders and Huber, 2010* | https://bioconductor.org/packages/release/bioc/html/DESeq2.html | |
| Other (Deposited Data) | R64-2-1 S288C sacCer3 genome assembly | Saccharomyces Genome Database (SGD) | https://downloads.yeastgenome.org/sequence/S288C_reference/genome_releases/ | |
| Other (Deposited Data) | Raw and analyzed data | This paper | GEO: GSE107841 | Contains raw and analyzed RNA-seq data |
| Other (Deposited Data) | Ribosomal profiling data | *Young et al., 2015* | GEO: GSE69414 | |
| Other (Deposited Data) | Codon protection index data | *Pelechano et al., 2015* | GEO: GSE63120 | |
| Other (Deposited Data) | Normalized yeast codon optimality scores | *Pechmann and Frydman, 2013* | http://www.stanford.edu/group/frydman/codons | |
| Other (Deposited Data) | Scaled estimates of yeast proteomic data | *Wang et al., 2012* | http://pax-db.org/ | |

## Yeast strains and plasmids

Yeast strains used in this study are in the W303 or BY4741 backgrounds and are listed in *Supplementary File 1*. The wild-type strain (HFY114) and its isogenic derivatives harboring deletions of *UPF1* (HFY871), *DCP1* (HFY1067), or *XRN1* (HFY1080) were described previously (*He et al., 2003*), as were isogenic strains harboring deletions of *DCP2* (CFY1016), *EDC3* (CFY25), *PAT1* (SYY2674), *LSM1* (SYY2680), or *DHH1* (SYY2686), or the *dcp2-N245* truncation of the Dcp2 C-terminal domain (SYY2385) and alleles thereof (*He and Jacobson, 2015a*). Isogenic strains harboring the C-terminally truncated, catalytically deficient *dcp2-E153Q-N245* (SYY2750) and *dcp2-E198Q-N245* (SYY2755) alleles or deletions of *SCD6* (SSY2352), *SKI2* (HFY1170), *SKI7* (SYY17), or both *SKI2* and *SKI7* (SYY21) were constructed by gene replacement (*Guthrie and Fink, 1991*) using DNA fragments harboring *dcp2-E153Q-N245::KanMX6*, *dcp2-E198Q-N245::KanMX6*, *scd6::KanMX6*, *ski2::URA3*, *ski7::URA3*, or *ski2::URA3* and *ski7::ADE2* null alleles, respectively. Double mutant strains *dcp2-N245 xrn1Δ* (SYY2887), *dcp2-E153Q-N245 xrn1Δ* (SYY2897), and *dcp2-E198Q-N245 xrn1Δ* (SYY2901) were

constructed by gene replacement using DNA fragments harboring the *xrn1::ADE2* null allele. Double mutant strains *dcp2-N245 ski2Δ* (SYY2889), *dcp2-N245 ski7Δ* (SYY2893) and *upf1Δ dhh1Δ* (SYY2700) were constructed by gene replacement using DNA fragments harboring *ski2::URA3*, *ski7::URA3*, and *dhh1::ADE2* null alleles, respectively. The use of temperature-sensitive *rpb1-1* cells to measure mRNA decay rates in yeast has been described previously (*Herrick et al., 1990*; *Dong et al., 2007*). Isogenic *rpb1-1* strains harboring deletions of *PAT1* (SYY2959), *LSM1* (SYY2965), and *DHH1* (SYY2971) were constructed by gene replacement using DNA fragments harboring *pat1::URA3*, *lsm1::URA3*, and *dhh1::URA3* null alleles, respectively. Genomic TAP-tagged *PAT1*, *LSM1*, and *DHH1* strains were obtained from Dharmacon, and isogenic TAP-tagged strains harboring deletions of *PAT1* (SYY2944 and SYY2950), *LSM1* (SYY2938 and SYY2953), and *DHH1* (SYY2941 and SYY2947) were constructed by gene replacement using DNA fragments harboring *pat1::URA3*, *lsm1::URA3*, and *dhh1::URA3* null alleles, respectively.

Yeast strains used for our RNA-Seq experiments have mostly been characterized in a previous study (*He and Jacobson, 2015a*). Doubling times of these strains in standard YEPD media at 30°C were: wild-type strain (HFY114), 1.5 hr; *dcp2-N245* strain (SYY2385), 1.8 hr; *dcp2-E153Q-N245* strain (SYY2750), 5.0 hr; *dcp2-E198Q-N245* strain (SYY2755), 3.6 hr; *pat1Δ* strain (SYY2674), 2.7 hr; *lsm1Δ* strain (SYY2680), 3.0 hr; and *dhh1Δ* strain (SYY2686), 2.3 hr.

Plasmids harboring the knock-in or knock-out alleles used in this study are described in *Supplementary File 2*.

## Cell growth and RNA isolation

Cells were all grown in YEPD media at 30°C. In each case, cells (15 ml) were grown to an $OD_{600}$ of 0.7 and harvested by centrifugation. Cell pellets were frozen on dry ice and then stored at −80°C until RNA isolation. Procedures for total RNA isolation and the measurement of mRNA decay rates using temperature-sensitive *rpb1-1* cells were as previously described (*He and Jacobson, 1995*).

## Analysis of protein levels

Preparation of whole-cell extracts and western blotting procedures were as described previously (*Dong et al., 2007*). Western blots were probed with specific antibodies, including a mouse mono-clonal antibody targeting the HA epitope (H3663, Sigma), rabbit polyclonal antibodies recognizing the TAP Tag (CAB1001, Thermo Fisher), and a mouse monoclonal antibody against Pgk1 (459250, Invitrogen). The latter polypeptide served as a loading control. Proteins were detected using ECL reagents (GE Healthcare) and Kodak BioMax film.

## RNA-Seq library preparation and sequencing

Procedures for RNA-Seq library construction were as previously described (*Celik et al., 2017*). In brief, total RNA was treated with Baseline-zero DNase (Epicenter) to remove any genomic DNA con-tamination. Five micrograms of DNase-treated total RNA was then depleted of rRNAs using the Illu-mina yeast RiboZero Removal Kit and the resulting RNA was used for RNA-Seq library preparation. Multiplex strand-specific cDNA libraries were constructed using the Illumina TruSeq Stranded mRNA LT Sample Prep Kit. Three independent cDNA libraries were prepared for each yeast strain analyzed. Total RNA cDNA libraries were sequenced on the Illumina HiSeq4000 platform at Beijing Genomics Institute. Four independent libraries were pooled into a single lane and single-end 50-cycle sequenc-ing was carried out for all cDNA libraries.

## Northern analysis

Procedures for northern blotting were as previously described (*He and Jacobson, 1995*). In each case, the blot was hybridized to a random primed probe for a specific transcript, with *SCR1* RNA serving as a loading control. Transcript-specific signals on northern blots were determined with a FUJI BAS-2500 analyzer. DNA fragments from the coding regions of specific genes were amplified by PCR using the oligonucleotides listed in *Supplementary File 3* and these DNA fragments were used as probes for the northern analyses. Probes generated for these analyses included *CIT2* nt 1–500, *SDS23* nt 1–500, *HOS2* nt 1–500, *PYK1* nt 1–500, *DIF1* nt 1–400, *AGA1* nt 1–500, *BUR6* nt 1–429, *LSM3* nt 1–270, *HXT6* nt 1–470, *GPH1* nt 1–500, *HXK1* nt 60–480, *CHA1* nt 1–500, *RTC3* nt 1–336, *NQM1* nt 481–1002, *PGM2* nt 1201–1710, *TMA10* nt 1–260, *GAD1* nt 1–480, *SPG4* nt 1–340,

*MUP3* nt 1–500, *GTT2* nt 1–500, *RPP1A* nt 1–321, *TMA19* nt 1–500, *GPP2* nt 1–500, *YIL164C* nt 201–600, *THI22* nt 1207–1700, *EST1* nt 1621–2094, *TRP1* nt 1–675, *ALR2* nt 1–500, *GDH1* nt 766–1365, *ARL1* nt 1–552, *DAL3* nt 1–588, *YGL117W* nt 95–691, *RPS9A* nt 599–1095, *SUC2* nt 1001–1599, *CPA1* nt 668–1236, *HIS4* nt 97–2328, *SER3* nt 811–1410, *HAC1* nt 662–913 and *HSP82* nt 1544–2130.

## Bioinformatic methods

### General computational methods

All statistical analyses were carried out using the R statistical programming environment, versions 3.3.2 and 3.3.4. R packages ggplot2, gplots, plyr, reshape2, and gridExtra were used for data pre-processing and visualization. Biostrings, BiocParallel, doSnow, and doParallel were used for parallel processing. Statistical tests were performed using built-in functions in base R distributions. Hierarchical clustering was performed using Euclidian distances between libraries and transcripts with complete linkage. Non-finite (division by 0) and undefined (0 divided by 0) values were removed prior to clustering. The heights of the clustering tree branches indicate distance between two libraries. We used Fisher's exact test to assess different subsets of transcripts for either enrichment or depletion of a particular group of transcripts. We used external data (codon protection index, codon optimality, and protein abundance) as presented by the respective authors without any further refinement. Transcripts that were not included in these datasets were discarded prior to statistical testing.

### Analysis of differential mRNA expression

Transcripts differentially expressed in each of the mutant strains relative to the corresponding wild-type strain were identified using bioinformatics pipelines described previously (*Celik et al., 2017*). In brief, the *Saccharomyces cerevisiae* R64-2-1 S288C reference genome assembly (sacCer3) was used to construct a yeast transcriptome comprised of 7473 transcripts. This transcriptome includes all annotated protein-coding sequences, functional and non-coding RNAs, and the unspliced isoforms of all intron-containing genes, but excludes all of the autonomous replicating sequences and long terminal repeats of transposable elements. The RSEM program (*Li and Dewey, 2011*) was used to map sequence reads to the transcriptome and to quantify the levels of individual mRNAs with settings –bowtie-m 30 –no-bam-output –forward-prob 0. The expected read counts for individual mRNAs from RSEM were considered as the number of reads mapped to each transcript and were then imported into the Bioconductor DESeq package (*Anders and Huber, 2010*) for differential expression analysis. The Benjamini-Hochberg procedure was used for multiple testing corrections. To account for replicate variability, we used a false discovery threshold of 0.01 (1%) instead of an arbitrary fold change cutoff as the criterion for differential expression.

### Analysis of potential mechanisms of mRNA decay

Our expression analysis identified the transcripts regulated by Pat1, Lsm1, and Dhh1. To assess the potential decay mechanisms for the respective sets of transcripts, we analyzed the expression patterns of these mRNAs in mutant cells deficient in decapping or 5' to 3' exoribonuclease activities. In addition, to assess the degree of co-translational decay, we also analyzed the codon protection indices (*Pelechano et al., 2015*) of these mRNAs in wild-type yeast cells under normal growth coditions. In our analyses, transcripts regulated by Pat1, Lsm1, and Dhh1 could be divided into six different subgroups. The up-regulated transcripts were grouped into Up-o-d, Up-o-pl, and Up-a-pld and the down-regulated transcripts were grouped into Down-o-d, Down-o-pl, and Down-a-pld subgroups (see Results). Boxplots were used to examine the distribution and the median value of both the relative levels and the codon protection indices for transcripts from each of these six subgroups. The relative levels of individual mRNAs in mutant cells were determined by comparison to their levels in wild-type cells. The expression data for *dcp1Δ*, *dcp2Δ*, and *xrn1Δ* cells were from our previously published work (*Celik et al., 2017*). Codon protection indices for individual mRNAs were generated by Pelechano and collegues based on their data from 5'P sequencing of yeast decay intermediates (*Pelechano et al., 2015*). In their study, the codon protection index of a specific transcript is defined as the ratio of sequencing reads in the ribosome protected frame over the average reads of the non-protected frames. Codon protection index values greater than one are indicative of co-translational decay.

## Analysis of intrinsic properties associated with mRNA translation

To assess the potential links between translation and the functions of Pat1, Lsm1, and Dhh1 in mRNA decay, we examined several intrinsic properties associated with mRNA translation for transcripts regulated by these three factors. We analyzed the pattern and distribution of the average codon optimality score, the average ribosome density, and the estimated protein abundance for transcripts from the six subgroups of mRNAs controlled by Pat1, Lsm1, and Dhh1. In our analysis, the average codon optimality score of individual mRNAs was calculated based on the optimal or non-optimal codon scores defined by Pechmann and Frydman (*Pechmann and Frydman, 2013*). The average ribosome density of individual mRNAs was derived from published ribosome profiling and RNA-Seq data from wild-type yeast cells grown under standard conditions (*Young et al., 2015*) and was calculated as previously described (*Celik et al., 2017*). In brief, raw fastq files were downloaded and sequence reads were trimmed for adapter sequences using cutadapt with settings -a CTGTAGGCA -q 10 –trim-n -m 10. After adapter trimming, sequence reads were mapped to the transcriptome using bowtie (*Langmead et al., 2009*) with settings -m4-n2l15 –suppress 1,6,7,8 –best –strata. After bowtie alignment, the riboSeqR (*He and Jacobson, 2015b*) was used for preliminary visualizations and frame calling. For our ribosome occupancy calculations, we selected read lenghts that showed a strong preference (> 80%) to a specific reading frame. After this filtering, the ribosome occupancy of individual mRNAs was calculated as coverage_ribo/coverage_rna, yielding a single value of ribosome occupancy for each mRNAs. We used these values to compare the translation efficiency of transcripts from different subgroups of mRNAs that are differentially expressed in *pat1Δ*, *lsm1Δ*, and *dhh1Δ* strains. We excluded transcripts that had no RNA-Seq reads mapping to their ORFs in our analysis. Protein abundance levels came directly from the curated PaxDb (Protein Abundances Across Organisms) database (*Wang et al., 2012*) and are the scaled aggregated estimates over several proteomic data sets.

## Deposited Data

The data discussed in this publication have been deposited in NCBI's Gene Expression Omnibus and are accessible through GEO Series accession number GSE107841 at the link https://www.ncbi.nlm.nih.gov/geo/query/acc.cgi?acc=GSE107841.

## Acknowledgements

This work was supported by grants to AJ (5R01 GM27757-37 and 1R35GM122468-01) from the US National Institutes of Health.

## Additional information

### Funding

| Funder | Grant reference number | Author |
| --- | --- | --- |
| National Institutes of Health | 5R01 GM27757-37 | Allan Jacobson |
| National Institutes of Health | 1R35GM122468- 01 | Allan Jacobson |

The funders had no role in study design, data collection and interpretation, or the decision to submit the work for publication.

### Author contributions

Feng He, Conceptualization, Resources, Data curation, Software, Formal analysis, Validation, Investigation, Visualization, Methodology, Writing—original draft, Writing—review and editing; Alper Celik, Conceptualization, Data curation, Software, Formal analysis, Validation, Investigation, Methodology, Writing—original draft, Writing—review and editing; Chan Wu, Formal analysis, Validation; Allan Jacobson, Conceptualization, Resources, Formal analysis, Supervision, Funding acquisition, Investigation, Methodology, Writing—original draft, Project administration, Writing—review and editing

## Author ORCIDs
Allan Jacobson  http://orcid.org/0000-0002-5661-3821

### Decision letter and Author response
Decision letter https://doi.org/10.7554/eLife.34409.030
Author response https://doi.org/10.7554/eLife.34409.031

## Additional files

### Supplementary files
• Supplementary File 1. Yeast strains used in this study.
DOI: https://doi.org/10.7554/eLife.34409.017

• Supplementary File 2. Plasmids used in this study.
DOI: https://doi.org/10.7554/eLife.34409.018

• Supplementary File 3. Oligonucleotides used in this study.
DOI: https://doi.org/10.7554/eLife.34409.019

• Transparent reporting form
DOI: https://doi.org/10.7554/eLife.34409.020

### Data availability
The data discussed in this publication have been deposited in NCBI's Gene Expression Omnibus and are accessible through GEO Series accession number GSE107841 at the link https://www.ncbi.nlm.nih.gov/geo/query/acc.cgi?acc=GSE107841.

The following dataset was generated:

| Author(s) | Year | Dataset title | Dataset URL | Database and Identifier |
| --- | --- | --- | --- | --- |
| He F, Celik A | 2017 | Genome-wide identification of decapping substrates in the yeast Saccharomyces cervisiae | https://www.ncbi.nlm.nih.gov/geo/query/acc.cgi?acc=GSE107841 | NCBI Gene Expression Omnibus, GSE107841 |

The following previously published datasets were used:

| Author(s) | Year | Dataset title | Dataset URL | Database and Identifier |
| --- | --- | --- | --- | --- |
| Young DJ, Guydosh NR, Zhang F, Hinnebusch AG, Green R | 2015 | Ribosome profiling study of rli1 depeletion strain | https://www.ncbi.nlm.nih.gov/geo/query/acc.cgi?acc=GSE69414 | NCBI Gene Expression Omnibus, GSE69414 |
| Pelechano V, Wei W, Steinmetz LM | 2015 | Widespread Co-translational RNA Decay Reveals Ribosome Dynamics | https://www.ncbi.nlm.nih.gov/geo/query/acc.cgi?acc=GSE63120 | NCBI Gene Expression Omnibus, GSE63120 |
| Celik A, Baker R, He F, Jacobson A | 2016 | A High Resolution Profile of NMD Substrates in Yeast | https://www.ncbi.nlm.nih.gov/geo/query/acc.cgi?acc=GSE86428 | NCBI Gene Expression Omnibus, GSE86428 |

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
