## [Decision Letter]

Thank you for submitting your article "General decapping activators target different subsets of inefficiently translated mRNAs" for consideration by *eLife*. Your article has been reviewed by three peer reviewers, and the evaluation has been overseen by a Reviewing Editor and James Manley as the Senior Editor.

The reviewers have discussed the reviews with one another and the Reviewing Editor has drafted this decision to help you prepare a revised submission.

Summary:

Jacobson and coworkers present an analysis by deep-sequencing of the transcriptome of various yeast strains carrying mutations in the decapping enzyme, Dcp2, and in some of its co-factors, particularly Dhh1, Pat1, and Lsm1. They show that the non-structured Dcp2 C-terminal domain has a regulatory function in mRNA decapping. importantly they demonstrate that Dhh1 and Pat1/Lsm1 decapping activators may target a selective set of genes. The data raise the possibility that different decay pathways, such as classical decapping activator-dependent, and NMD pathways may compete for the Dcp1/Dcp2 decapping complex.

Essential revisions:

Your paper was reviewed by three experts in the field. It is clear that the paper addresses a very important question, which is the substrate specificity of the different decapping activators in the regulation of mRNA levels in yeast cells. The finding that decapping activators selectively promote the reduction of specific mRNAs is important and novel. However, as you can see from the reviewers’ comments, the paper suffers from several deficiencies, of which the most problematic one deals with the interpretation of the data concerning RNA stability. It is obvious that steady-state RNA levels cannot inform on RNA stability, and therefore without knowing the mechanism of specificity of decapping one can invoke several scenarios, as elaborated by reviewer #3. For example, Dhh1 and Pat1/Lsm1 apparent preference of different targets could just reflect different mechanistic impacts of removing factors having different activities, such as Xrn1.

We therefore request that you perform new experiments and change the text to respond to the reviewers' criticisms as follows:

1) Because steady-state RNA levels do not necessarily reflect RNA stability, direct validation of a subset of each group of targets for RNA stability is necessary.

2) You need to monitor factor levels. Since many decapping activators are known to interact and possibly stabilize each other, the data should be presented that deletion of a specific activator has no effect on levels of other activators. This information could be also available in the literature. In addition, you need to provide a Western of the different Dcp2 truncated proteins and take into account any substantial differences in expression.

3) The yeast mutants should be better characterized.

4) The presentation should be improved to fully account for the literature (see below).

5) You speculate that some of the observations could be explained by the 3' to 5' mRNA degradation pathway being activated or unlocked (for example, subsection “Identification of transcripts uniquely and commonly targeted by Pat1, Lsm1, and Dhh1”, second paragraph). You should test the appropriate *ski2* or *ski7* mutant strains if they are available.

*Reviewer #1:*

A bioinformatics approach was used to analyze the genome-wide transcriptome of yeast strains harboring a truncation of the Dcp2 decapping enzyme C-terminal regulatory domain and the regulators of decapping, PAT1, LSM1, or DHH1. Several key novel findings are reported. First, removal of the Dcp2 C-terminus, unleashes the Dcp2 catalytic domain to indiscriminately target RNAs for rapid decay of transcripts that it would otherwise not degrade, supporting the modulatory function of the C-terminus on decapping. RNA-Seq data from the Dcp2 C-terminal truncated allele (*dcp2-N245*) showed a large number of C-terminal responsive transcripts upregulated, supporting the previously reported autoinhibitor regulatory function in the C-terminus. Second, the decapping activator proteins, thought to be generic stimulators of Dcp2 decapping, are instead transcript specific and selectively target mRNAs to Dcp2 decapping. This is an important new finding and reveals an additional layer of transcript specific regulation for what appears to be the primary decapping enzyme in yeast. Transcripts that are both direct and indirect decapping targets are identified and further bioinformatically characterized to correlate to their translatability. An impressive array of Northern blots is used throughout the manuscript to validate steady state bioinformatic data. Overall the data considerably expands our mechanistic understanding of Dcp2 decapping and the substrate specificity of the respective decapping activators in regulation of mRNA levels in yeast cells.

One minor clarification should be incorporated: The observed reduction in transcript levels in the decapping protein or decapping activator proteins are interpreted to be exclusively a function of mRNA turnover throughout the manuscript. Since only steady state values are measured, changes in transcription rates/efficiency should also be discussed.

*Reviewer #2:*

Two main findings reported in this work are:

a) Extension of previous observations from Jacobson's lab that the non-structured Dcp2 C-terminal domain has a regulatory function in mRNA decapping (this is supported by extensive RNA seq and biochemical, analyses of a battery of different yeast strains), and;

b) Demonstration that Dhh1 and Pat1/Lsm1 decapping activators may target specific set of genes. The latter finding is well documented by the sequencing and Northern data performed with a dozen or so different strains. An interesting, though limited, set of experiments is also suggesting that different decay pathways depending on mRNA decapping, such as classical decapping activator dependent pathways and NMD may compete for the Dcp1/Dcp2 decapping complex.

Generally, the experiments are well designed and conclusive, and the results should be of general interest and also stimulate future biochemical studies on a role of individual decapping activators in decay of specific targets. This is certainly not a well-understood issue at present. Description of specific role of individual decapping activators in regulation of different classes of mRNA is a novel interesting finding. Below are some other more specific comments:

1) In a number of places in Results and Discussion the authors speculate that some of the observations could be explained by the 3' to 5' mRNA degradation pathway being activated or unlocked (for example, subsection “Identification of transcripts uniquely and commonly targeted by Pat1, Lsm1, and Dhh1”, second paragraph). The authors should test (or construct) appropriate *ski2* or *ski7* mutant strains to document this speculation.

2) Since many decapping activators are known to interact with each other and possibly stabilize each other, the data should be presented that deletion of a specific activator has no effect on levels of other activators. Information on that should be probably also available in the literature.

3) Regarding competition of the NMD and regular decapping pathways (subsection “Deletion of DHH1 promotes the degradation of a fraction of NMD substrates” and subsection “Pat1, Lsm1, and Dhh1 also have indirect roles in controlling genome-wide mRNA expression”, last paragraph), are there any data in the literature that Dcp1/Dcp2 complex may be limiting under some conditions? Any evidence of its level or activity being regulated by growth conditions?

4) When discussing physical interaction between Dhh1 and CCR4 /Not (subsection “Pat1, Lsm1, and Dhh1 target subsets of yeast transcripts with overlapping substrate specificity”, second paragraph and subsection “General computational methods”), refer also to papers by Chen et al., 2014 (Izaurralde's group) and Rouya et al., 2014 (Sonenberg's group).

*Reviewer #3:*

In this manuscript, Jacobson and coworkers present an analysis by deep-sequencing of the transcriptome of various yeast strains carrying mutations in the decapping enzyme, Dcp2, and in some of its co-factors, particularly Dhh1, Pat1 and Lsm1. Based on these results, in part validated by Northern blots, they suggest that various decapping co-factors activate the decay of specific subsets of yeast mRNAs.

The main weaknesses of this manuscript are:

1) The interpretation of mRNA levels determined by deep-sequencing as indicative of alteration of mRNA decay (for mRNAs up-regulated in decay mutants);

2) The absence of molecular evidence explaining the proposed specificity of RNA decay regulators for specific subsets of yeast mRNAs;

3) The poor characterization of the yeast mutants analyzed, leaving opportunities for alternative interpretations of the data;

4) The incomplete presentation and analysis of the literature.

These points are detailed below.

1) In this manuscript the authors interpret increased mRNA levels in mutant versus wild-type yeast as an indication that mRNA decay is altered (e.g., "up-regulated transcripts are most likely bona fide substrates of the yeast decapping enzyme"). At the same time, they are surprised by the fact that levels of some mRNAs are decreased between mutant and wild-type yeasts (subsection “Elimination of the large Dcp2 C-terminal domain causes significant changes in genome-wide mRNA expression”, second paragraph) and suggests repeatedly that changes in mRNA decay may induce secondary effects. This reasoning is not convincing: up-regulation or down-regulation of transcript levels are as likely to be indirect effects (e.g., up-regulation may result from the down-regulation of a repressor). The authors' conclusions are thus not sufficiently solid.

Moreover, studies by others have shown that analyses of mRNA levels don't allow strong conclusions about transcripts whose stability have been altered. Indeed, changes in mRNA decay may be compensated by concomitant modification of transcription rates (e.g., Sun et al., Genome Res. 2012 Jul;22(7):1350-9). To circumvent this problem various strategies assessing the mRNA transcription and decay rates have been implemented over the last 5 years. The use of mRNA levels determined by sequencing as indicative of changes in mRNA decay is currently not appropriate and doesn't allow to derive strong conclusions. An alternative method of analysis should have been used. Altogether, the data and conclusions reported in this manuscript don't seem to meet the requirement for "highest scientific standards and importance" necessary for publication in *eLife*.

2) As a consequence of their analysis (without considering the severe limitation above) the authors conclude that Pat1/Lsm1 and Dhh1 target different subsets of mRNAs. This conclusion remains, however, descriptive. One would expect the authors to provide a molecular mechanism explaining the specificity of at least one of these factors.

3) Authors compare different mutant yeast strains that are likely to behave very differently, at least with respect to growth rates. This parameter is not mentioned. Yet, if growth-rates are different, changes in transcript levels will automatically ensue as the proportion of cells in different stages of the cell cycle will vary. This could account for many indirect up-regulation or down-regulation of transcript levels. This parameter should minimally have been discussed.

Similarly, authors assume that truncation of Dcp2 to residue 245 result in constitutive activation of the protein. Yet, truncation of Dcp2 may well stabilize the short resulting protein. This possibility is consistent with the observation that the C-terminal domain of Dcp2 is target of ubiquitination (Swaney DL et al., 2013, PMID: 23749301). Some of the data presented may thus need to be interpreted in a context of an increase of the decapping enzyme level, not as resulting from its constitutive activation. Similar issues apply to the other Dcp2 mutants. Authors should minimally have monitored levels of the wild type and mutant proteins to allow strong conclusions.

4) Available literature is incompletely presented and cited, for example:

- It is surprising that the widely conserved Edc1 decapping activator is not mentioned in the manuscript!

- Subsection “Dcp2 C-terminal domain imparts critical in vivo regulatory activities in mRNA decapping”, second paragraph, authors indicate that most biochemical structural studies of Dcp2 focus on the 1-245 fragments quoting studies that have in some case targeted larger proteins. Some recent work is also not quoted (e.g., Mugridge et al., 2018).

- Introduction fails to quote recent work and often focus on the authors work. For example, recent structural analyses have provided insights into the role of the Dcp2 N-terminal domain in decapping but authors only quote structural work of 2008 (Introduction, first paragraph). Also, authors argue that Edc3 interacts with "the large C-terminal domain of Dcp2" when recent studies have shown that it activates Dcp2 in part by binding its N-terminal region.

- Discussion of differences between Dhh1 and Pat1 fails to mention that the latter has been shown to directly recruit Xrn1 (Charenton et al., 2017). This fact could explain differences in the steady state levels of some transcripts as, in contrast to Dhh1, absence of Pat1 may block both decapping and the downstream degradation of the mRNA body that will be monitored by deep-sequencing. Thus, the conclusion of the authors suggesting that Dhh1 and Pat1/Lsm1 may target different subsets of yeast mRNAs may simply reflect the different mechanistic impacts of removing factors having different activities etc.

Many minor points would also need to be improved and this manuscript suffers mostly from strong weaknesses. Overall, I believe that this manuscript doesn't meet the requirement for "highest scientific standards and importance" necessary for publication in *eLife*.

---

## [Author Response]

Essential revisions:Your paper was reviewed by three experts in the field. It is clear that the paper addresses a very important question, which is the substrate specificity of the different decapping activators in the regulation of mRNA levels in yeast cells. The finding that decapping activators selectively promote the reduction of specific mRNAs is important and novel. However, as you can see from the reviewers’ comments, the paper suffers from several deficiencies, of which the most problematic one deals with the interpretation of the data concerning RNA stability. It is obvious that steady-state RNA levels cannot inform on RNA stability, and therefore without knowing the mechanism of specificity of decapping one can invoke several scenarios, as elaborated by reviewer #3. For example, Dhh1 and Pat1/Lsm1 apparent preference of different targets could just reflect different mechanistic impacts of removing factors having different activities, such as Xrn1.We therefore request that you perform new experiments and change the text to respond to the reviewers' criticisms as follows:1) Because steady-state RNA levels do not necessarily reflect RNA stability, direct validation of a subset of each group of targets for RNA stability is necessary.

We carried out the recommended experiments and present their results in new figures (Figures 6 and B). Our approach utilized temperature shifts in *rpb1-1* strains to inactivate transcription, a method that we and others have used extensively in the past. New double mutants (*pat1Δ/rpb1-1, lsm1Δ/rpb1-1,* and *dhh1Δ/rpb1-1*) were constructed, and half-lives were determined for 14 mRNAs representative of the upregulated Up-o-d, Up-o-pl, and Up-a-pld subgroups (see subsection “Validation of transcripts controlled directly or indirectly by Pat1, Lsm1, and Dhh1”). As we now note in the text, the results of these experiments “[…] indicate that the increased steady-state accumulation of transcripts from the three subgroups in *pat1*∆, *lsm1*∆, and *dhh1*∆ cells are most likely direct consequences of the loss of Pat1, Lsm1, and Dhh1 functions in mRNA decapping and suggest that at least a significant fraction of transcripts from these three subgroups are direct targets of Pat1, Lsm1, and Dhh1.”

2) You need to monitor factor levels. Since many decapping activators are known to interact and possibly stabilize each other, the data should be presented that deletion of a specific activator has no effect on levels of other activators. This information could be also available in the literature. In addition, you need to provide a Western of the different Dcp2 truncated proteins and take into account any substantial differences in expression.

We carried out the recommended experiments and present their results in new figures (Figure 1—figure supplement 4 and Figure 3—figure supplement 2). Our approach utilized western blotting of extracts from cells expressing: a) triple-HA tagged *dcp2-N245, dcp2E153Q-N245*, and *dcp2-E198Q-N245* alleles of *DCP2* in otherwise wild-type backgrounds for mRNA decay factors or b) commercially available TAP-tagged alleles of *PAT1, LSM1,* and *DHH1* in strains systematically lacking each of the decapping activators under study. Details of these experiments can be found in the last paragraph of the subsection “Elimination of the large Dcp2 C-terminal domain causes significant changes in genome-wide mRNA expression” and in the last paragraph of the subsection “Decapping activators Pat1, Lsm1, and Dhh1 target specific subsets of yeast transcripts with overlapping substrate specificity”. The relevant methodology has been added in the subsection “Yeast strains and plasmids”. These experiments demonstrated that: a) the levels of the Dcp2 proteins expressed in *dcp2-N245, dcp2-E153Q-N245*, and *dcp2-E198Q-N245* cells were similar or nearly identical (Figure 1—figure supplement 4) and b) deletion of *PAT1* had little or no effect on the levels of expression of Lsm1 and Dhh1 (Figure 3—figure supplement 2B and C), deletion of *DHH1* had no significant effect on the levels of expression of Pat1 and Lsm1 (Figure 3—figure supplement 2A and B) and deletion of *LSM1* had no effect on the level expression of Dhh1 (Figure 3—figure supplement 2C). Consistent with a previous observation (Bonnerot et al., 2000), deletion of *LSM1* decreased the level of expression of Pat1 to 35% of its level in wild-type cells (Figure 3—figure supplement 2A). The decreased accumulation of Pat1 in the absence of Lsm1 does not contradict our overall interpretation of overlapping mRNA expression patterns between *pat1*∆ and *lsm1*∆ cells, and, in fact, this observation strengthens our conclusion that Pat1 and Lsm1 function together to promote mRNA decapping.

3) The yeast mutants should be better characterized.

The Materials and methods section has been revised to include information on the origin, construction, and growth rates of the different yeast mutants used in our study.

4) The presentation should be improved to fully account for the literature (see below).

Six additional references have been added to the Introduction and Discussion. They are highlighted in these sections, and in the bibliography.

5) You speculate that some of the observations could be explained by the 3' to 5' mRNA degradation pathway being activated or unlocked (for example, subsection “Identification of transcripts uniquely and commonly targeted by Pat1, Lsm1, and Dhh1”, second paragraph). You should test the appropriate ski2 or ski7 mutant strains if they are available.

We constructed *pat1Δ/ski2Δ, lsm1Δ/ski2Δ,* and *dhh1Δ/ski2Δ* double mutants and used northern blotting to assess whether 3’ to 5’ decay might be responsible for diminished levels of some mRNAs in *pat1Δ, lsm1Δ,* and *dhh1Δ* mutants. The results of these experiments, presented in Figure 4—figure supplement 1, showed that deletion of *SKI2* partially increased the levels of the *GTT2* and *RPP1A* mRNAs in *pat1∆* and *lsm1∆* cells, but did not increase the levels of these two mRNAs in *dhh1∆* cells. These results are now noted in the last paragraph of the subsection “Identification of transcripts uniquely and commonly targeted by Pat1, Lsm1, and Dhh1”. We also attempted to construct double mutants of *dcp2Δ, dcp2-E153Q-N245, dcp2-E198Q-N245, or xrn1Δ* with *ski2Δ or ski7Δ* to further test the proposed decay mechanism. However, we were unable to generate even a single double mutant as all combinations were lethal in our strain background, an observation seen previously in published data from the Parker lab (Anderson and Parker, 1998).

Reviewer #1:[…] One minor clarification should be incorporated: The observed reduction in transcript levels in the decapping protein or decapping activator proteins are interpreted to be exclusively a function of mRNA turnover throughout the manuscript. Since only steady state values are measured, changes in transcription rates/efficiency should also be discussed.

As noted in our response to “essential revision” #1 (above), this concern has been addressed with new experiments and additional figures and text.

Reviewer #2:[…] Generally, the experiments are well designed and conclusive, and the results should be of general interest and also stimulate future biochemical studies on a role of individual decapping activators in decay of specific targets. This is certainly not a well-understood issue at present. Description of specific role of individual decapping activators in regulation of different classes of mRNA is a novel interesting finding. Below are some other more specific comments:1) In a number of places in Results and Discussion the authors speculate that some of the observations could be explained by the 3' to 5' mRNA degradation pathway being activated or unlocked (for example, subsection “Identification of transcripts uniquely and commonly targeted by Pat1, Lsm1, and Dhh1”, second paragraph). The authors should test (or construct) appropriate ski2 or ski7 mutant strains to document this speculation.

As noted in our response to “essential revision” #5 (above), this concern has been addressed with new experiments and an additional figure and additional text.

2) Since many decapping activators are known to interact with each other and possibly stabilize each other, the data should be presented that deletion of a specific activator has no effect on levels of other activators. Information on that should be probably also available in the literature.

As noted in our response to “essential revision” #2 (above), this concern has been addressed with new experiments and additional figures and text.

3) Regarding competition of the NMD and regular decapping pathways (subsection “Deletion of DHH1 promotes the degradation of a fraction of NMD substrates” and subsection “Pat1, Lsm1, and Dhh1 also have indirect roles in controlling genome-wide mRNA expression”, last paragraph), are there any data in the literature that Dcp1/Dcp2 complex may be limiting under some conditions? Any evidence of its level or activity being regulated by growth conditions?

We are not aware any published data suggesting that the Dcp1/Dcp2 complex may be limiting under some conditions. We mined the SGD database and found that even under normal growth conditions, the number of Dcp1 or Dcp2 molecules per cell are fewer than the sum of the number of molecules of the Dcp2-interacting decapping activators including Upf1, Pat1, Edc3 and Scd6, suggesting that the Dcp1/Dcp2 decapping enzyme is limiting for the formation of different decapping complexes. Work from the Parker lab indicated that Dcp2 is phosphorylated under several stress conditions and one specific phosphorylation event appeared to be linked to the stability of a subset of mRNAs (Yoon et al., 2010), suggesting that cellular decapping activity is likely regulated by growth conditions. To take these observations into account we have added a sentence (subsection “Deletion of DHH1 promotes the degradation of a fraction of NMD substrates”, last paragraph) acknowledging that the decapping enzyme may be limiting under normal growth conditions.

4) When discussing physical interaction between Dhh1 and CCR4 /Not (subsection “Pat1, Lsm1, and Dhh1 target subsets of yeast transcripts with overlapping substrate specificity”, second paragraph and subsection “General computational methods”), refer also to papers by Chen et al., 2014 (Izaurralde's group) and Rouya et al., 2014 (Sonenberg's group).

As noted in our response to “essential revision” #4 (above), these references have been added to the text.

Reviewer #3:[…] 1) In this manuscript the authors interpret increased mRNA levels in mutant versus wild-type yeast as an indication that mRNA decay is altered (e.g., "up-regulated transcripts are most likely bona fide substrates of the yeast decapping enzyme"). At the same time, they are surprised by the fact that levels of some mRNAs are decreased between mutant and wild-type yeasts (subsection “Elimination of the large Dcp2 C-terminal domain causes significant changes in genome-wide mRNA expression”, second paragraph) and suggests repeatedly that changes in mRNA decay may induce secondary effects. This reasoning is not convincing: up-regulation or down-regulation of transcript levels are as likely to be indirect effects (e.g., up-regulation may result from the down-regulation of a repressor). The authors' conclusions are thus not sufficiently solid.Moreover, studies by others have shown that analyses of mRNA levels don't allow strong conclusions about transcripts whose stability have been altered. Indeed, changes in mRNA decay may be compensated by concomitant modification of transcription rates (e.g., Sun et al., Genome Res. 2012 Jul;22(7):1350-9). To circumvent this problem various strategies assessing the mRNA transcription and decay rates have been implemented over the last 5 years. The use of mRNA levels determined by sequencing as indicative of changes in mRNA decay is currently not appropriate and doesn't allow to derive strong conclusions. An alternative method of analysis should have been used. Altogether, the data and conclusions reported in this manuscript don't seem to meet the requirement for "highest scientific standards and importance" necessary for publication in eLife.

As noted in our response to “essential revisions #1” (above), we have addressed this concern by determining half-lives for mRNAs representative of the three upregulated subgroups of mRNAs. We are aware of genome wide approaches to measuring mRNA decay rates, but find these methods to still be problematic because of the poor correlations between different studies (Geisberg et al., 2014, Grigull et al., 2004, Holstege et al., 1998, Miller et al., 2011, Munchel et al., 2011, Presnyak et al., 2015, Wang et al., 2002).

2) As a consequence of their analysis (without considering the severe limitation above) the authors conclude that Pat1/Lsm1 and Dhh1 target different subsets of mRNAs. This conclusion remains, however, descriptive. One would expect the authors to provide a molecular mechanism explaining the specificity of at least one of these factors.

We (and apparently reviewers 1 and 2) consider the identification and translational phenotypes of substrates targeted by the different decapping activators to be a significant advance in the field. While understanding the molecular mechanisms underlying the observed specificity would be fascinating, we consider that to be a long-term goal of future studies.

3) Authors compare different mutant yeast strains that are likely to behave very differently, at least with respect to growth rates. This parameter is not mentioned. Yet, if growth-rates are different, changes in transcript levels will automatically ensue as the proportion of cells in different stages of the cell cycle will vary. This could account for many indirect up-regulation or down-regulation of transcript levels. This parameter should minimally have been discussed.Similarly, authors assume that truncation of Dcp2 to residue 245 result in constitutive activation of the protein. Yet, truncation of Dcp2 may well stabilize the short resulting protein. This possibility is consistent with the observation that the C-terminal domain of Dcp2 is target of ubiquitination (Swaney DL et al., 2013, PMID: 23749301). Some of the data presented may thus need to be interpreted in a context of an increase of the decapping enzyme level, not as resulting from its constitutive activation. Similar issues apply to the other Dcp2 mutants. Authors should minimally have monitored levels of the wild type and mutant proteins to allow strong conclusions.

As noted in our response to “essential revisions #3’ (above), we have included additional information in the Materials and methods that addresses strain origins, construction, and growth rates. It’s worth noting that the doubling times of the *pat1∆, lsm1∆,* and *dhh1∆* strains (2.7 hr, 3.0 hr, and 2.3 hr, respectively) do not differ to extents likely to affect the conclusions of our studies.

As for truncation of Dcp2 to amino acid 245, two points should be noted: a) constitutive activation of this truncated form of Dcp2 was established in our previous study by systematic deletion analysis accompanied by assessments of protein level and activity (He et al., 2015) and b) as noted in our response to “essential revisions #2” (above), we have determined the levels of the proteins expressed by the different *dcp2* alleles and shown that they are markedly consistent.

4) Available literature is incompletely presented and cited, for example:- It is surprising that the widely conserved Edc1 decapping activator is not mentioned in the manuscript!

Edc1 is widely conserved, and a short peptide of this protein has been characterized in several biochemical studies, but the evidence for its in vivo function in mRNA decapping is, in our opinion, still weak. That said, we acknowledge that the reviewer may have insights that we’re unaware of and have added four Edc1 references to the Introduction (first and second paragraphs).

- Subsection “Dcp2 C-terminal domain imparts critical in vivo regulatory activities in mRNA decapping”, second paragraph, authors indicate that most biochemical structural studies of Dcp2 focus on the 1-245 fragments quoting studies that have in some case targeted larger proteins. Some recent work is also not quoted (e.g., Mugridge et al., 2018).

We were careful to add the qualifying word “largely” to the sentence referred to by the reviewer. Of the papers cited in this section only one study used a slightly larger Dcp2 fragment Dcp2 (1-275 amino acids). We included that study in our list of pertinent papers because we have previously shown that the first 300 amino acids of yeast Dcp2 has constitutive decapping activity in vivo (He et al., 2015).

The Mugridge et al., 2018 reference has been added to the manuscript (Introduction, first paragraph and subsection “Dcp2 C-terminal domain imparts critical in vivo regulatory activities in mRNA decapping”, last paragraph).

- Introduction fails to quote recent work and often focus on the authors work. For example, recent structural analyses have provided insights into the role of the Dcp2 N-terminal domain in decapping but authors only quote structural work of 2008 (Introduction, first paragraph). Also, authors argue that Edc3 interacts with "the large C-terminal domain of Dcp2" when recent studies have shown that it activates Dcp2 in part by binding its N-terminal region.

As noted in multiple places above we have added several new references to the paper that address additional decapping studies from other labs. However, our interpretation of the nature of the Edc3 binding site on Dcp2 differs from that of the reviewer. We have consistently defined the “N-terminal” region of Dcp2 as amino acids 1-245 (see Discussion). Since the paper alluded to by the reviewer (Charenton et al., 2016) found an additional Edc3 binding site downstream of amino acids 1-245 we chose not to consider this as an Nterminal binding site.

- Discussion of differences between Dhh1 and Pat1 fails to mention that the latter has been shown to directly recruit Xrn1 (Charenton et al., 2017). This fact could explain differences in the steady state levels of some transcripts as, in contrast to Dhh1, absence of Pat1 may block both decapping and the downstream degradation of the mRNA body that will be monitored by deep-sequencing. Thus, the conclusion of the authors suggesting that Dhh1 and Pat1/Lsm1 may target different subsets of yeast mRNAs may simply reflect the different mechanistic impacts of removing factors having different activities etc.

We now discuss the potential role of Pat1 in 5’ to 3’ exonucleolytic decay (subsection “Pat1, Lsm1, and Dhh1 target subsets of yeast transcripts with overlapping substrate specificity”, second paragraph), but we have a different perspective on the possible role of such interactions in the regulation of mRNA decay. The reviewer’s model predicts that *pat1∆* and *xrn1∆* cells would have similar phenotypes, which they don’t. As noted in our response to this reviewer’s point 2 (above), it’s too soon to get precise about mechanism without much more data so we are hesitant to speculate about the role of the Pat1:Xrn1 interaction or the lack of an apparent Dhh1:Xrn1 interaction.

In short, we believe that the results of our paper shed considerable new light on cellular decapping regulation and we certainly hope that you and the reviewers consider the revised manuscript to be suitable for publication in *eLife*.